# Profilin 1 deficiency drives mitotic defects and reduces genome stability

Federica Scotto di Carlo [1], Sharon Russo [1,2], Francesc Muyas[3], Maria Mangini [4], Lorenza Garribba[5], Laura Pazzaglia[6], Rita Genesio[7], Flavia Biamonte [8,9], Anna Chiara De Luca [4], Stefano Santaguida [5,10], Katia Scotlandi [6], Isidro Cortés-Ciriano [3✉] & Fernando Gianfrancesco [1✉]

Profilin 1—encoded by *PFN1*—is a small actin-binding protein with a tumour suppressive role in various adenocarcinomas and pagetic osteosarcomas. However, its contribution to tumour development is not fully understood. Using fix and live cell imaging, we report that Profilin 1 inactivation results in multiple mitotic defects, manifested prominently by anaphase bridges, multipolar spindles, misaligned and lagging chromosomes, and cytokinesis failures. Accordingly, next-generation sequencing technologies highlighted that Profilin 1 knock-out cells display extensive copy-number alterations, which are associated with complex genome rearrangements and chromothripsis events in primary pagetic osteosarcomas with Profilin 1 inactivation. Mechanistically, we show that Profilin 1 is recruited to the spindle midzone at anaphase, and its deficiency reduces the supply of actin filaments to the cleavage furrow during cytokinesis. The mitotic defects are also observed in mouse embryonic fibroblasts and mesenchymal cells deriving from a newly generated knock-in mouse model harbouring a *Pfn1* loss-of-function mutation. Furthermore, nuclear atypia is also detected in histological sections of mutant femurs. Thus, our results indicate that Profilin 1 has a role in regulating cell division, and its inactivation triggers mitotic defects, one of the major mechanisms through which tumour cells acquire chromosomal instability.

[1] Institute of Genetics and Biophysics "Adriano Buzzati-Traverso" (IGB), National Research Council of Italy (CNR), Naples, Italy. [2] Department of Environmental, Biological and Pharmaceutical Sciences and Technologies (DiSTABiF), University of Campania Luigi Vanvitelli, Caserta, Italy. [3] European Molecular Biology Laboratory, European Bioinformatics Institute, Wellcome Genome Campus, Hinxton, UK. [4] Institute for Experimental Endocrinology and Oncology, "G. Salvatore" (IEOS), National Research Council of Italy (CNR), Naples, Italy. [5] Department of Experimental Oncology at IEO, European Institute of Oncology IRCCS, Milan, Italy. [6] IRCCS Istituto Ortopedico Rizzoli, Laboratory of Experimental Oncology, Bologna, Italy. [7] Department of Molecular Medicine and Medical Biotechnology, University of Naples Federico II, Naples, Italy. [8] Department of Experimental and Clinical Medicine, Magna Graecia University, Catanzaro, Italy. [9] Center of Interdepartmental Services (CIS), Magna Graecia University, Catanzaro, Italy. [10] Department of Oncology and Hemato-Oncology, University of Milan, Milan, Italy. ✉email: icortes@ebi.ac.uk; fernando.gianfrancesco@igb.cnr.it

Correct chromosome segregation is a fundamental requirement for eukaryotic organisms to maintain genome stability through mitotic divisions[1,2]. The cytoskeleton plays a major role in ensuring faithful chromosome segregation, providing both structural support to maintain cell shape and the mitotic spindle for the movement of the dividing chromosomes[3]. Errors during cell division might generate changes in chromosome content, leading to aneuploidies and polyploidies[4]. Although most aneuploid karyotypes arising during early embryonic life are lethal, aneuploidy occurring in adulthood has been linked to aging and tumorigenesis[1,5]. The ongoing acquisition of alterations in chromosome number and structure is defined as chromosomal instability (CIN), a hallmark of human cancers that contributes to increasing intra-tumour heterogeneity and genomic instability[6–8]. Structural nuclear abnormalities, such as chromosome bridges[9–11] and micronuclei[12], are considered morphological markers of CIN[7,13] and are associated with defective mitosis. However, our understanding of the molecular mechanisms underpinning aberrant mitosis remains incomplete.

While chromosome alterations are detected in almost all malignant tumours, their prevalence varies across cancers[14,15]. Osteosarcoma (OS)—the most common cancer of bone[16]—represents one of the tumours with the highest degree of genomic instability[17,18]. It mainly affects children and adolescents, where it constitutes a remarkable cause of cancer-related death[16,19–21]. In elderly patients (>60 years), OS is often secondary to Paget's disease of bone (OS/PDB), and usually presents with a poor prognosis[22,23]. Cancer genome studies revealed that OS are predominantly characterised by a high burden of structural variants (SVs) and complex genome rearrangements[17,18,24,25]. Therefore, recurrent alterations in OS are not frequently detected and clear cancer drivers are not identified. Some molecular understanding about OS development can be gained through well-defined genetic syndromes that show increased risk of OS: germline pathogenic variants have been identified in *TP53* (Li-Fraumeni syndrome), *RB1* (retinoblastoma), *RECQL4* (Rothmund–Thomson syndrome), *WRN* (Werner syndrome), and *BLM* (Bloom syndrome)[19]. We recently identified a loss-of-function mutation in the *PFN1* gene (c.318_321del) in a hereditary form of OS arising in Paget's disease of bone, and somatic *PFN1* loss of heterozygosity (LOH) was also detected in sporadic OS/PDB patients[26], thus suggesting a potential role for Profilin 1 in OS pathogenesis. *PFN1* encodes Profilin 1, the most studied member of a conserved family of four small actin-binding proteins involved in the organisation of the cellular membrane and cytoskeleton[27,28]. Profilin 1 mediates the dynamic remodelling of the actin cytoskeleton by recharging actin monomers with ATP for new filament assembly[29]. *Pfn1*-null mice are embryonically lethal at the two-cell stage, indicating that Profilin 1 is essential for cell division and survival[30]. In contrast, Profilin 1 is downregulated in various adenocarcinomas (breast[31–33], hepatic[34], pancreatic[35,36], and bladder[37]), which correlates with increased metastatic potential and shorter overall survival[36], suggesting a dosage-dependent effect on a fine-tuned balance between cellular lethality and neoplastic transformation. Although low levels of Profilin 1 have been associated with disruption of cytokinesis and impairment of the contractile ring in both *Drosophila melanogaster*[38] and mouse[39] models, whether this could represent the underlying mechanism for tumour onset has not been investigated. Accurate coordination of the assembly/disassembly of the actin cytoskeleton is a crucial requirement to shape cells, especially to guide mitosis[3,40]. We therefore hypothesised that the reduction of Profilin 1 observed in OS and adenocarcinomas underpins tumour onset due to mitotic errors.

In this study, we show that reduced Profilin 1 levels drive mitotic defects that foster CIN, as indicated by the frequent formation of micronuclei, chromosome bridges, and lagging and misaligned chromosomes. Profilin 1 deficiency results in extensive somatic copy-number alterations (SCNAs) in a mesenchymal cellular model. In addition, genome sequencing of OS/PDB tumour biopsies with *PFN1* loss of heterozygosity shows complex genome rearrangements, including chromothripsis. Together, our results show that reduced Profilin 1 levels correlate with increased cell division defects that promote genomic instability.

## Results

### Profilin 1 localises to the spindle midzone to promote actin accumulation at the cleavage furrow.
To determine the role of Profilin 1 in cell division, we analysed its subcellular localisation during different stages of mitosis. For this purpose, we used human RPE1 cells, which are diploid, non-transformed, hTERT-immortalised retinal pigment epithelial cells widely used as a model to study mitosis[41–45]. Wild type (WT) RPE1 cells were synchronised, immunostained and analysed throughout mitosis, from prophase to near completion of cytokinesis (Methods section). Profilin 1 initially showed a uniform distribution throughout the cytoplasm, and then accumulated between the condensed chromosomes in metaphase (Fig. 1). In anaphase, an intense signal at the central region of the spindle became visible, indicating strong recruitment of Profilin 1 from the cytoplasmic pool to the spindle midzone, which decreased in late anaphase and telophase (Fig. 1). The mitotic localisation of Profilin 1 was also analysed in three heterozygous and three homozygous *PFN1* knock-out (KO) RPE1 clones, which we generated using CRISPR/Cas9 (Supplementary Fig. 1a, b; Methods section). *PFN1*$^{+/-}$ RPE1 clones manifested a general reduction of Profilin 1 expression, and lacked an overt enrichment of Profilin 1 in the midzone; whereas *PFN1*$^{-/-}$ cells were expectedly devoid of Profilin 1 in the cytoplasm (Fig. 2a). To examine whether the accumulation of Profilin 1 in the midzone was a feature unique to RPE1 cells or rather shared by other cell types, the immunostaining was repeated on additional murine and human non-transformed cell lines, i.e., MC3T3, dermal fibroblast, and HK-2. This assay confirmed that Profilin 1 was detected at the midzone area also in these cell contexts, indicating a universal role for this protein in this region during cell division (Supplementary Fig. 2). The spindle midzone is required to properly position the cleavage furrow, whose ingression is mediated by contractile forces generated by filaments of actin and myosin[46,47]. Therefore, we investigated the impairment of the contractile ring in *PFN1*-KO cells by examining actin nucleation at the cleavage furrow of *PFN1*$^{+/-}$ and *PFN1*$^{-/-}$ RPE1 cells through phalloidin staining. Of note, the fluorescence intensity of F-actin within the cleavage furrow of *PFN1*$^{+/-}$ and *PFN1*$^{-/-}$ cells was markedly decreased compared with WT RPE1 ($P = 0.0004$ and $P < 0.0001$, respectively; one-way ANOVA), demonstrating the relevance of Profilin 1 in regulating actin polymerisation at specific stages of mitosis (Fig. 2b, c).

### Lack of Profilin 1 triggers prolonged mitosis due to impaired mitotic entry and exit.
To determine the effect of Profilin 1 deficiency on mitosis, we used time-lapse phase-contrast light microscopy (Methods section). We followed fields of WT and KO RPE1 cells for 15–20 h after thymidine synchronisation at early S phase and subsequent release (Fig. 3a). We observed that, while $65.3 \pm 4.8\%$ of WT cells committed to mitosis after ~5/6 h following release, only $13.0 \pm 6.5\%$ of *PFN1*$^{+/-}$ and $24.7 \pm 3.8\%$ of *PFN1*$^{-/-}$ RPE1 entered mitosis within ~10/12 h from release (Fig. 3b). In addition, while normal cells took 18.5 min (95% Confidence Interval (CI): 17.5–19.4) from mitotic rounding to cytokinesis, KO cells spent 30.1 min (95% CI: 26.1–34.0) in mitosis before completing cell division (Fig. 3c–e and Supplementary Movies 1–3). We also observed cases in which mutant cells underwent cytokinesis failure

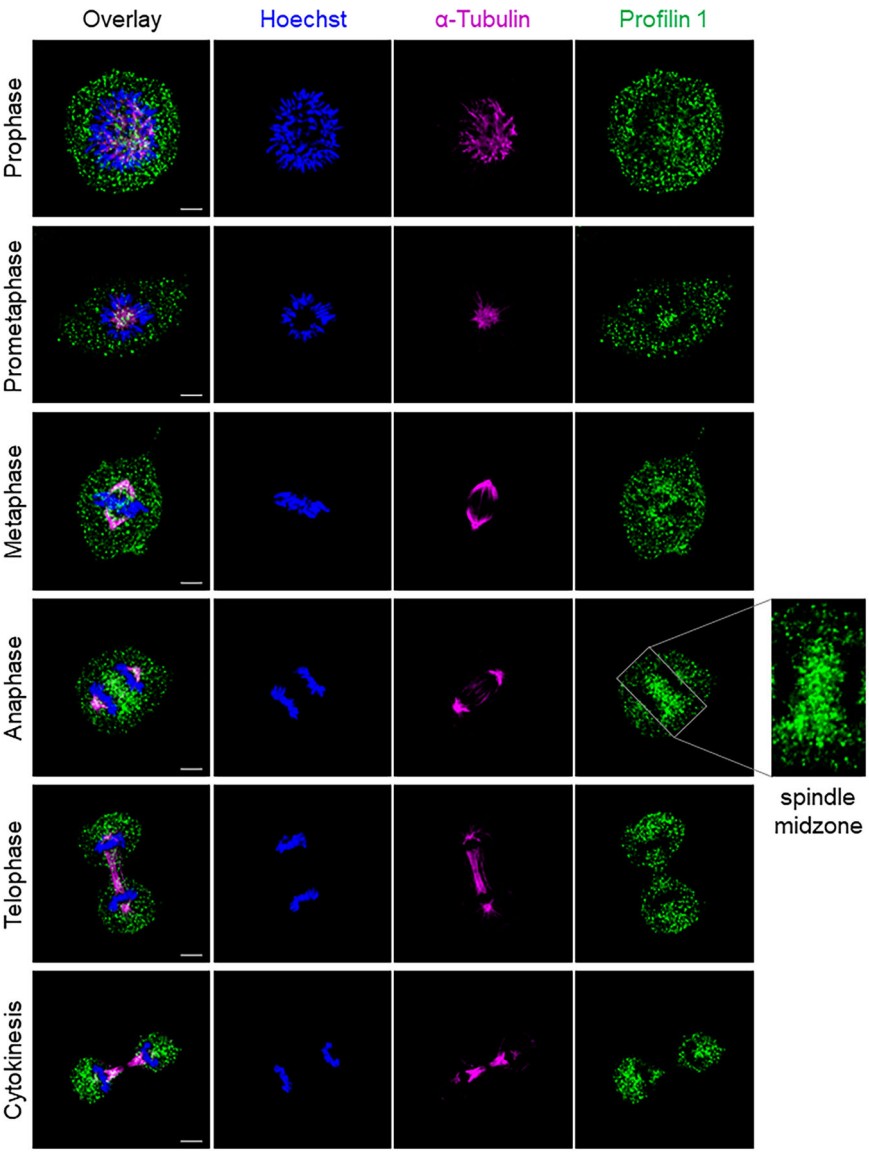

**Fig. 1 Profilin 1 localisation during the mitotic stages.** Immunofluorescent staining for the mitotic spindle (α-tubulin, magenta) and Profilin 1 (green) in normal RPE1 cells during mitosis; DNA is shown in blue (Hoechst 33342); scale bars 5 μm.

(5 of 66 mitoses of $PFN1^{+/-}$ and 3 of 42 mitoses of $PFN1^{-/-}$ RPE1), resulting in a doubled genome in the subsequent interphase, which was never detected in 127 mitoses of WT cells (Supplementary Fig. 3a, b). These results are in agreement with the lower amount of actin at the contractile ring in $PFN1$ mutant cells (Fig. 2b, c). Upon transition to mitosis, cells reduce their adhesions to the substrate and round up through the formation of a dense cortical actin network[3]. Interestingly, we observed that $PFN1$-KO cells were able to disassemble focal adhesions only partially, acquired an incomplete spherical shape and failed to completely round up. Some cells ended up succeeding cell division, yet most cells took several attempts to initiate the M phase without effectively dividing during the 15–20 h of time-lapse observation (Supplementary Fig. 3c and Supplementary Movies 1–3). The difficulty of KO cells to acquire a spherical shape let us speculate that the lack of cell rounding could be the consequence of insufficient levels of cortical actin[48,49]. Fluorescent staining of F-actin revealed that the intense circular band of actin in control mitotic cells was decreased in $PFN1^{+/-}$ ($P = 0.0002$, one-way ANOVA) and $PFN1^{-/-}$ cells ($P < 0.0001$, one-way ANOVA; Fig. 3f, g). Of note, we observed a remarkable difference in cell shape in metaphase, with KO RPE1 unable to achieve a complete rounding

and exhibiting an irregular shape, whereas rounded metaphase cells were found in control cultures (Fig. 3f). Because in vivo mitotic cells generate protrusive forces that physically deform the surrounding collagen fibres of the extracellular matrix to allow for mitotic rounding and elongation[40,50,51], we suspected that defective rounding might be rescued by growth on collagen plates. However, growth on collagen-coated coverslips failed to restore the spherical geometry of $PFN1$-KO mitotic cells (Supplementary Fig. 4a), indicating that the observed effect is not substrate-dependent. These data demonstrate that Profilin 1 loss dramatically harms the morphology of mitotic cells, compromising mitotic rounding and cytokinesis, and delaying the overall cell division process. Surprisingly, stable overexpression of Profilin 1 in $PFN1^{+/-}$ and $PFN1^{-/-}$ RPE1 cells failed to rescue a normal mitotic duration. On the contrary, its overexpression delayed mitotic progression in WT cells, which spent an average of 27.1 min (95% CI: 25.9 – 28.3) in mitosis versus 18.5 min of parental WT cells, suggesting that excess in actin nucleation likewise results in prolonged mitosis (Supplementary Fig. 4b). This result is in agreement with activating mutations in other actin-promoting proteins[52–54], which lead to delayed mitoses. Accordingly, stable overexpression of the actin monomer-binding Profilin 1

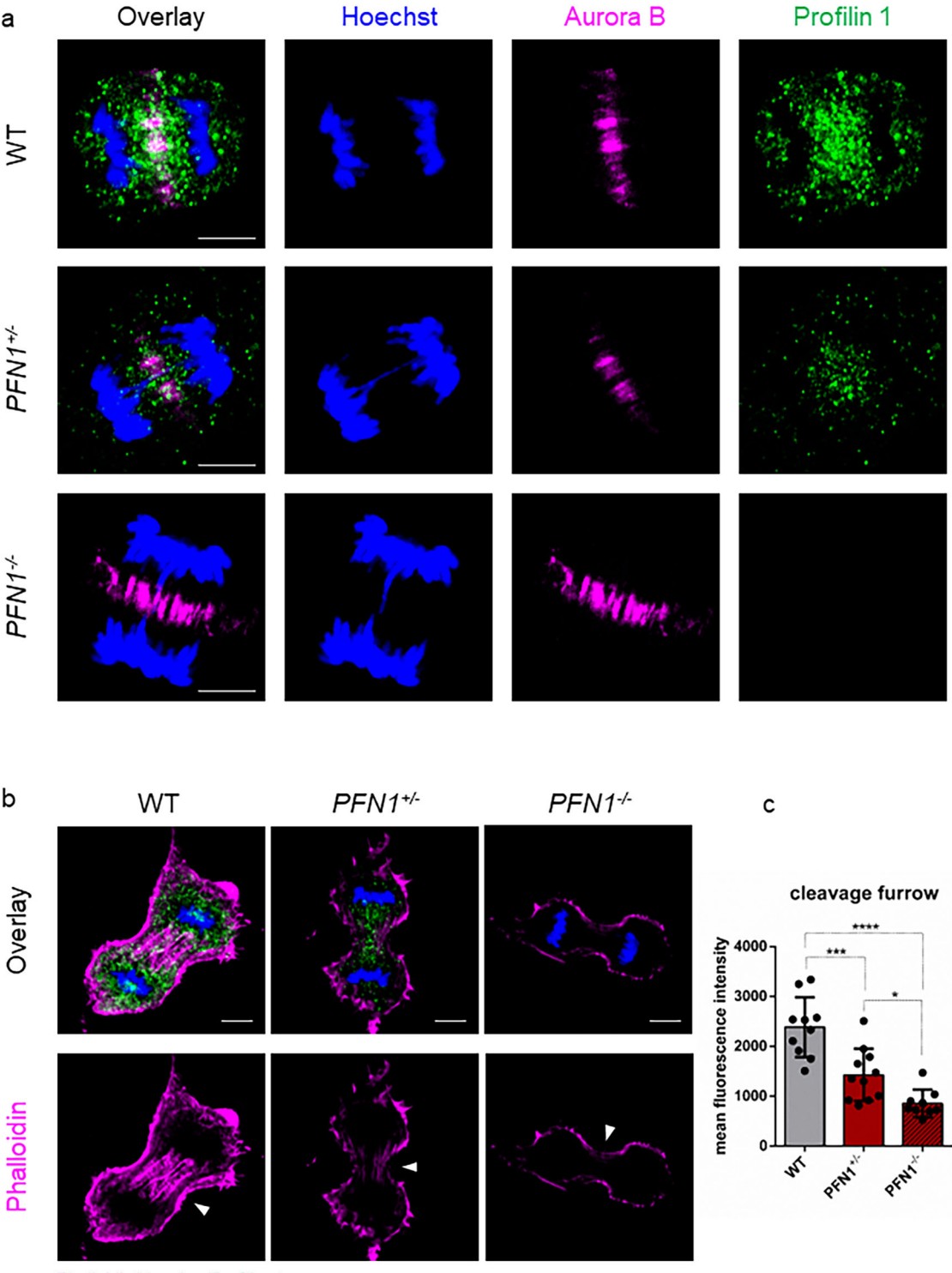

**Fig. 2 Profilin 1 localises in the mitotic midzone and promotes actin polymerisation at the cleavage furrow. a** WT, *PFN1*[+/−] and *PFN1*[−/−] RPE1 cells at anaphase stained for DNA (Hoechst 33342, blue), a midzone marker (Aurora B kinase, magenta) and Profilin 1 (green); scale bars 5 μm. Note that *PFN1*-KO cells show normal Aurora B localisation. **b** WT, *PFN1*[+/−] and *PFN1*[−/−] RPE1 cells at telophase stained for DNA (Hoechst 33342, blue), Profilin 1 (green) and F-actin (phalloidin, magenta). Arrowheads point towards furrow ingression; scale bars 5 μm. **c** Mean fluorescence intensity of phallodin staining at the cleavage furrow; data are shown as mean ± s.e.m.; dots represent the fluorescent intensity for each sample (*n* = 10 for WT, 11 for *PFN1*[+/−], 9 for *PFN1*[−/−] cells); \*\*\**P* = 0.0004, \*\*\*\**P* < 0.0001, \**P* = 0.0461. Ordinary one-way ANOVA performed.

mutant (PFN1[R89E]; ref. [55]) was not associated with mitotic delay in WT cells, and did not worsen the delayed phenotype in KO cells. Thus, balanced levels of Profilin 1, and therefore of actin, are necessary to modulate the kinetics of mitosis and to ensure a timely mitotic execution.

**Profilin 1-dependent mitotic delay is accompanied by chromosome segregation defects.** To investigate the effect of Profilin 1 loss on chromosome segregation, we determined the fraction of mitotic WT and *PFN1*-KO RPE1 cells with chromosome mis-segregation events after thymidine synchronisation and release

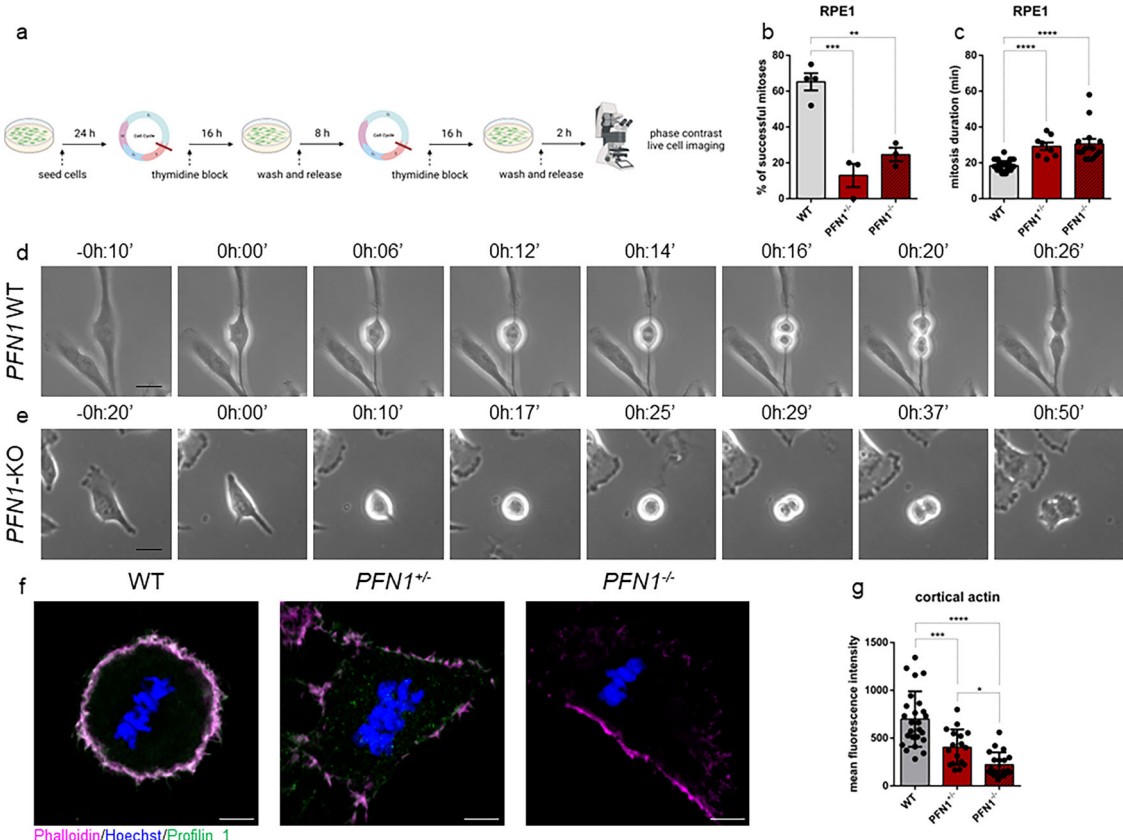

**Fig. 3 Profilin 1 knock-out induces prolonged mitosis. a** Schematic of the experiment created in BioRender (https://biorender.com/) by F.S.d.C. as a registered user. **b** Percentage of cells able to complete mitosis following the round up; data are shown as mean ± s.e.m.; dots represent the mean for each experiment (*n* = 67 WT, 44 *PFN1*⁺/⁻, 43 *PFN1*⁻/⁻ RPE1 cells; range of number of cells used for each individual experiment: 9-25 WT, 13-16 *PFN1*⁺/⁻, 11-16 *PFN1*⁻/⁻); ***P = 0.0005, **P = 0.0023. The mean difference between *PFN1*⁺/⁻ and *PFN1*⁻/⁻ cells was not statistically significant (P = 0.4310). **c** Time taken from start (cell rounding) to end (cytokinesis) of mitosis; data are shown as mean ± s.e.m.; dots represent the duration of each cell division expressed in minutes (*n* mitoses = 39 WT, 8 *PFN1*⁺/⁻, 14 *PFN1*⁻/⁻ RPE1 cells); ****P < 0.0001. Data in **b** and **c** were analysed by ordinary one-way ANOVA. **d** Representative time-lapse phase-contrast images (taken every 2 min) of WT and **e** *PFN1*-KO RPE1 cells; scale bars 25 μm. Time stamps indicate elapsed time in hours:minutes. **f** WT, *PFN1*⁺/⁻ and *PFN1*⁻/⁻ RPE1 cells at metaphase stained for DNA (Hoechst 33342, blue), Profilin 1 (green) and F-actin (phalloidin, magenta); scale bars 5 μm. **g** Mean fluorescence intensity of cortical actin; data are shown as mean ± s.e.m.; dots represent the fluorescent intensity for each sample (*n* = 25 for WT, 17 for *PFN1*⁺/⁻, 20 for *PFN1*⁻/⁻ cells); ***P = 0.0002, ****P < 0.0001, *P = 0.0436. Ordinary one-way ANOVA performed.

through confocal microscopy analysis of cells with immunostained microtubules (Fig. 4a and Methods section). Both *PFN1*⁺/⁻ and *PFN1*⁻/⁻ RPE1 cells displayed chromosome segregation defects at higher frequency than the parental cell line (Fig. 4b–i and Supplementary Table 1). Remarkably, a higher frequency of chromosome misalignments at the metaphase plate was observed in *PFN1*⁺/⁻ (P = 0.0096, one-way ANOVA) and *PFN1*⁻/⁻ RPE1 cells (P = 0.0411, one-way ANOVA) compared to WT (Fig. 4b, f). Normal spindle assembly was also disrupted with increased formation of multipolarity in *PFN1*⁺/⁻ (P = 0.0305, one-way ANOVA) and *PFN1*⁻/⁻ cells (P = 0.0259, one-way ANOVA; Fig. 4c, g). Furthermore, evaluation of microscopy images of anaphases and telophases revealed that *PFN1*⁺/⁻ and *PFN1*⁻/⁻ cells were more prone to chromosome bridges (P = 0.0003 and P = 0.0009, respectively; one-way ANOVA) (Fig. 4d, h and Supplementary Fig. 4c) and contained lagging chromosomes at a higher frequency than control cells (P = 0.2457 and P = 0.0345, respectively; one-way ANOVA) (Fig. 4e, i). The frequency of nuclear atypia detected in homozygous KO clones and heterozygous clones was similar (P > 0.05, Student's *t* test), indicating that the loss of one *PFN1* functional copy is sufficient to induce CIN (Supplementary Table 1). Together, these results show that Profilin 1 insufficiency destabilises

the regulation of mitosis, leading to chromosome segregation abnormalities during cell division.

To rule out the possibility that the formation of abnormal nuclear structures could be an unintended off-target effect of CRISPR/Cas9[56,57], we analysed mitotic RPE1 cells stably expressing either non-target or *PFN1*-shRNA (Supplementary Fig. 5a, b; Methods section). Anaphase bridges were the most common defects, and the knockdown of *PFN1* triggered their formation at a higher frequency as compared to cells expressing the control shRNA (P = 0.0041, Student's *t* test) (Supplementary Fig. 5c,d). Additionally, 3.0 ± 0.9% of anaphase/telophase cells contained lagging chromosome fragments and, rarely (2 out of 307), metaphase plates showed misaligned chromosomes (Supplementary Fig. 5e, f), which were not found in control cells (*n* = 183).

To follow the fate of daughter cells with mis-segregated chromosomes, we performed fluorescence live cell imaging of WT, *PFN1*⁺/⁻ and *PFN1*⁻/⁻ RPE1 cells stably expressing H2B-mCherry/EGFP-Tubulin (Methods section). The elevated occurrence of mitotic aberrations was confirmed in *PFN1*⁺/⁻ and *PFN1*⁻/⁻ RPE1 cells, namely chromosome misalignment on metaphase plates, anaphase bridges, micronuclei formation, and cytokinesis failures (Supplementary Fig. 6). Although we noted floating apoptotic cells in KO cultures, these mitotic errors

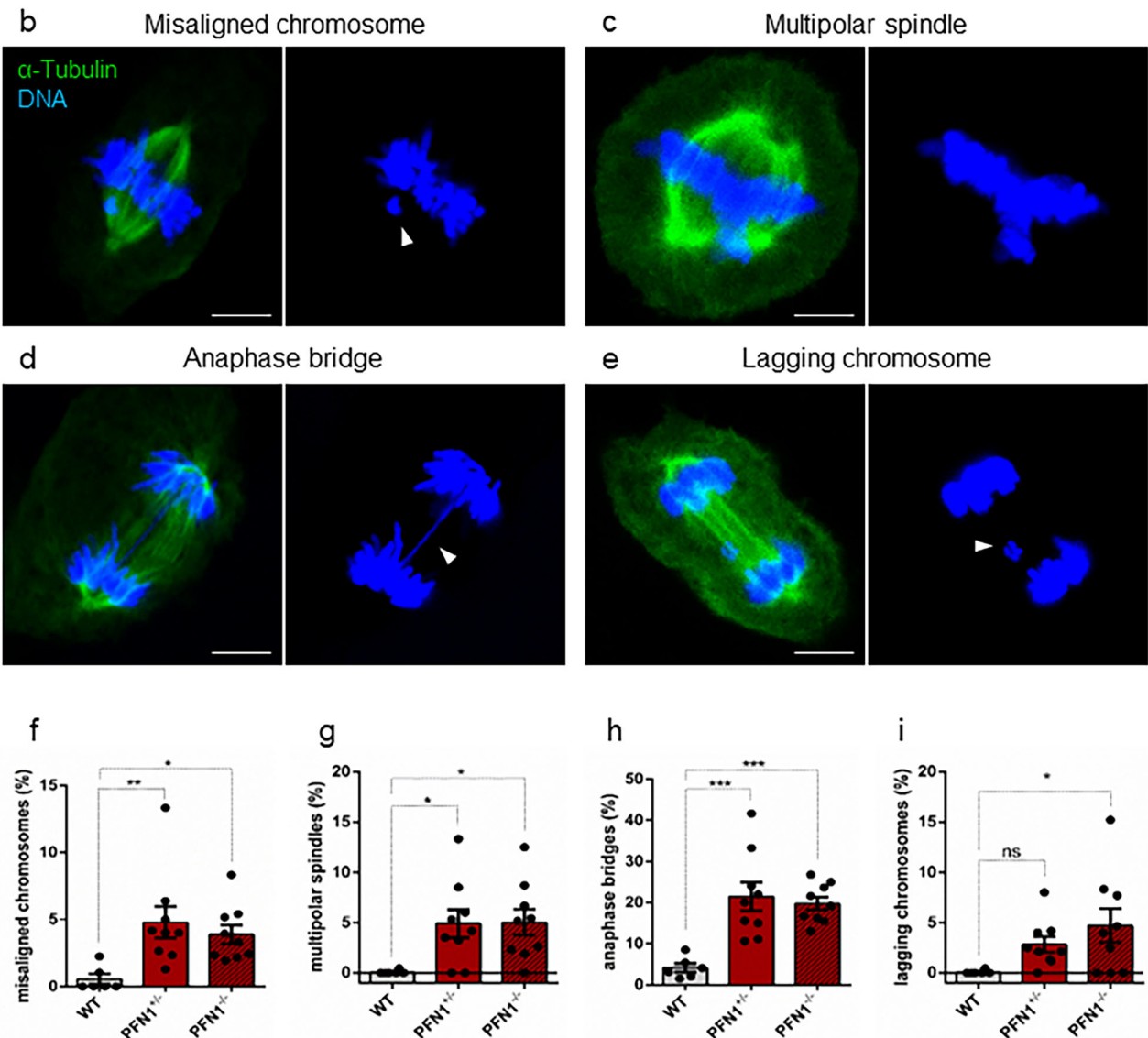

**Fig. 4 Chromosome segregation defects in Profilin 1-deficient RPE1 cells. a** Schematic of the experiment created in BioRender (https://biorender.com/) by F.S.d.C. as a registered user. Below, representative immunofluorescence images of mitotic *PFN1*-KO RPE1 cells showing **b** chromosome misalignment (arrowhead) at the metaphase plate, **c** multipolar spindle formation, **d** a chromosome bridge (arrowhead) persisting in late anaphase, **e** telophase with a lagging chromosome (arrowhead). Cells were stained for microtubules (α-tubulin, green) and DNA (Hoechst 33342, blue); scale bars 5 µm. In each panel, the overlay shows α-tubulin/DNA staining. Control metaphase, anaphase, and telophase images of dividing RPE1 cells are illustrated in Fig. 1.
**f** Quantification (%) of cells with misaligned chromosomes; **\*\****P* = 0.0096, *\*P* = 0.0411. **g** Quantification (%) of cells with multipolar spindles; *\*P* = 0.0305, *\*P* = 0.0259. **h** Quantification (%) of cells with anaphase bridges; **\*\*\****P* = 0.0003, **\*\*\****P* = 0.0009. **i** Quantification (%) of cells with lagging chromosomes; ns *P* = 0.2457, *\*P* = 0.0345. Data in **f–i** are shown as mean ± s.e.m. of three independent experiments (*n* = 2 biological replicates for WT, 3 for *PFN1*⁺ᐟ⁻, 3 for *PFN1*⁻ᐟ⁻ cells); dots represent the value of each experiment (scoring 911, 825 and 616 total mitoses in WT, *PFN1*⁺ᐟ⁻ and *PFN1*⁻ᐟ⁻ samples, respectively); range of number of cells used for each individual experiment: 70-266 WT, 30-189 *PFN1*⁺ᐟ⁻, 24-116 *PFN1*⁻ᐟ⁻. Ordinary one-way ANOVA performed.

resulted in viable daughter cells in most cases, suggesting that a p53-deficient background could enhance mitotic dysfunction and tumorigenicity[58–61] (Supplementary Movies 4–7).

**Profilin 1 insufficiency prompts mitotic errors in mesenchymal cells and in vivo.** Loss of function of *PFN1* was previously detected in OS/PDB[26], an aggressive tumour of mesenchymal origin. Therefore, to investigate the role of Profilin 1 in the maintenance of genome integrity also in mesenchymal cells, we analysed *Pfn1*-KO MC3T3 mouse cells that we engineered to obtain heterozygous KO clones[26]. We confirmed the impairment of cell division as a consequence of Profilin 1 loss, since $50.7 \pm 5.0\%$ of WT MC3T3 cells completed mitosis in the time-frame of observation *versus* $30.5 \pm 4.9\%$ of *Pfn1*[+/−] MC3T3 cells (Supplementary Fig. 7a, b). Mutant cells took on average 30.8 min (95% CI: 28.4–33.1) to divide, whereas control cells spent an average of 24.5 min (95% CI: 22.8–26.1) in mitosis (Supplementary Fig. 7c–e and Supplementary Movies 8 and 9). We then evaluated the rate of nuclear abnormalities in *Pfn1*[+/−] MC3T3 as compared to WT MC3T3 cells. Fluorescent staining of interphase nuclei revealed an increased frequency of nuclear abnormalities in mutant cells, namely binucleation ($P = 0.0004$, Student's *t* test; Fig. 5a, d), chromosome bridges ($P = 0.0021$, Student's *t* test) (Fig. 5b, e), and spontaneous formation of micronuclei ($P = 0.0049$, Student's *t* test) (Fig. 5c, f). To assess whether micronucleation was related to DNA damage, we measured the damage-dependent phosphorylation of the histone variant H2AX (γ-H2AX). We identified γ-H2AX positive-micronuclei in $67.2 \pm 5.6\%$ of *Pfn1*[+/−] cells, indicating that mutant micronuclei harboured DNA double strand breaks (DSBs; Fig. 5g, h). By contrast, only a few micronucleated WT cells ($6.3 \pm 2.4\%$) were positive for γ-H2AX (Fig. 5h). In the absence of cell stress, p53 is maintained at low levels; however, in the presence of DNA damage, it rapidly accumulates within the cell nucleus[62,63]. Therefore, to test whether the DSBs observed in *Pfn1*[+/−] cells activated a p53-mediated response, we measured p53 levels in WT and *Pfn1*[+/−] MC3T3 cells. We found that p53 levels increased concomitantly with the reduction of Profilin 1 (Fig. 5i, j). As expected, p53 accumulation was only detected in the nuclei of *Pfn1*[+/−] MC3T3 cells, especially in the cells showing micronuclei and chromosome bridges (Fig. 5k). Notably, p53 accumulation was also observed in *PFN1*[−/−] RPE1 cells, thus confirming that chronic Profilin 1 inactivation drives p53 expression (Supplementary Fig. 8). However, in RPE1 cells stably expressing a doxycycline-inducible shRNA (Methods section), p53 was undetectable after 7 days of *PFN1* silencing, indicating that acute reduction of Profilin 1 by 90% does not lead to p53 accumulation (Supplementary Fig. 8). Intriguingly, a constant silencing of Profilin 1 for 21 days— that is, mimicking chronic inactivation—triggered a modest p53 accumulation (Supplementary Fig. 8). These data demonstrate that p53 activation is not a direct and immediate consequence of Pro-filin 1 reduction, but rather derives from long-term Profilin 1 inactivation, perhaps as a result of accumulating errors.

To assess the consequences of Profilin 1 inactivation in vivo, we generated a mouse model harbouring the *Pfn1* loss-of-function mutation previously identified in OS/PDB (c.318_321del)[26] (Supplementary Fig. 9; Methods section). Homozygous *Pfn1* knock-in mice died in utero; therefore, we isolated mouse embryonic fibroblasts (MEFs) at 14.5 days post coitum in an attempt to track cell proliferation in WT and mutant cells. However, homozygous (*Pfn1*[c.318_321del/ c.318_321del]) embryos were not found even at this stage ($n = 44$ embryos), nor did females present resorbed decidua in the uterus, suggesting that *Pfn1* is essential for the earliest cell divisions in mouse embryos. Time-lapse phase-contrast microscopy of control (*Pfn1*[WT/WT]) and

heterozygous knock-in (*Pfn1*[c.318_321del/WT]) MEFs confirmed a high rate of mitotic defects in mutant cells, especially failures in mitotic rounding and cytokinesis (Supplementary Movies 10 and 11). We next validated this finding in mesenchymal cells isolated from calvarias of viable heterozygous newborns. Confocal analysis of nuclei revealed that *Pfn1*[c.318_321del/WT] mesenchymal cells showed an increased frequency of micronuclei formation as compared to *Pfn1*[WT/WT] cells ($P = 0.0312$, Student's *t* test; Fig. 6a, b). Although the frequency of binucleation was not significantly higher in *Pfn1*[c.318_321del/WT] cells compared with WT ($P = 0.0778$, Student's *t*-test) (Fig. 6c), we occasionally observed knock-in cells with three nuclei, which were never observed in control cell cultures (Fig. 6d). These results suggest that mitotic defects were more penetrant in the presence of the heterozygous loss-of-function mutation.

Given the frequent occurrence of OS in the epiphysis of long bones[16], we examined the growth plate of femurs of 4-month-old mice for the presence of aberrant mitoses. Interestingly, the analysis of the histological sections highlighted a 5-fold increase in the frequency of abnormally binucleated cells ($P = 0.0054$, Student's *t* test) and rare trinucleated cells ($P = 0.0532$, Student's *t* test) in Profilin 1-deficient growth plates compared with cells in femurs from WT mice (Fig. 6e–g), which is in line with the observed defective cytokinesis in *PFN1*-KO RPE1 cells (Supplementary Figs. 3 and 6) and in *Pfn1*[c.318_321del/WT] MEFs (Supplementary Movie 11). Collectively, these results indicate that reduced levels of Profilin 1 are sufficient to induce mitotic errors that result in nuclear abnormalities also in vivo.

**High frequency of genomic rearrangements in cellular models and pagetic osteosarcoma.** Osteosarcoma is a mesenchymal tumour characterised by extensive somatic copy-number aberrations (SCNAs) and SVs[17,18,24]. To test whether the nuclear abnormalities related to *Pfn1* haploinsufficiency effectively resulted in chromosome copy-number changes, we performed low-pass whole-genome sequencing (WGS) on 8 different single-cell-derived *Pfn1*[+/−] MC3T3 clones and the WT culture (Methods section). We first confirmed that all MC3T3 clones were efficiently knocked out through Western blot, Sanger sequencing analysis, and WGS (Supplementary Fig. 10a, b). Copy-number analysis revealed that all *Pfn1*[+/−] MC3T3 clones harboured at least one chromosomal alteration compared to WT MC3T3 cells (range: 2–13; Supplementary Fig. 11). This result suggests that the loss of *Pfn1* results in replication defects and consequent accumulation of SCNAs.

We next analysed human tumours to assess whether complex patterns of genome instability, such as chromothripsis, are present in OS/PDB genomes. To this aim, we performed WGS on paired tumour and matched normal samples for 4 primary OS tumours from PDB patients as well as whole-exome sequencing (WES) for 10 additional tumours (mean age at diagnosis 66 years old; 95% CI: 62–71; Fig. 7; Supplementary Data 1; and Methods section). The 4 WGS samples had polyploid genomes (ploidy ≥ 2.5) and a high number of SCNAs and SVs (range 202-324 SVs) (Fig. 7a and Supplementary Data 1), including chromothripsis affecting a single (chromosome 8 in patient 1798; Fig. 7a) or multiple chromosomes (e.g., chromosomes 5 and 13 in patient 1363) accompanied by high-level copy-number amplifications. Extensive loss of heterozygosity (LOH) across multiple chromosomes was frequently observed (Fig. 7b, c), and we detected LOH of *PFN1* in 3 out of 4 WGS samples, and in 7 out of 10 WES samples (Supplementary Data 1).

Copy-number analysis revealed that 8/14 tumours had undergone whole-genome doubling (WGD) events, with 3 samples undergoing WGD twice (Fig. 7d; Methods section). Given that multiple copies of the same parental *PFN1* allele are observed in

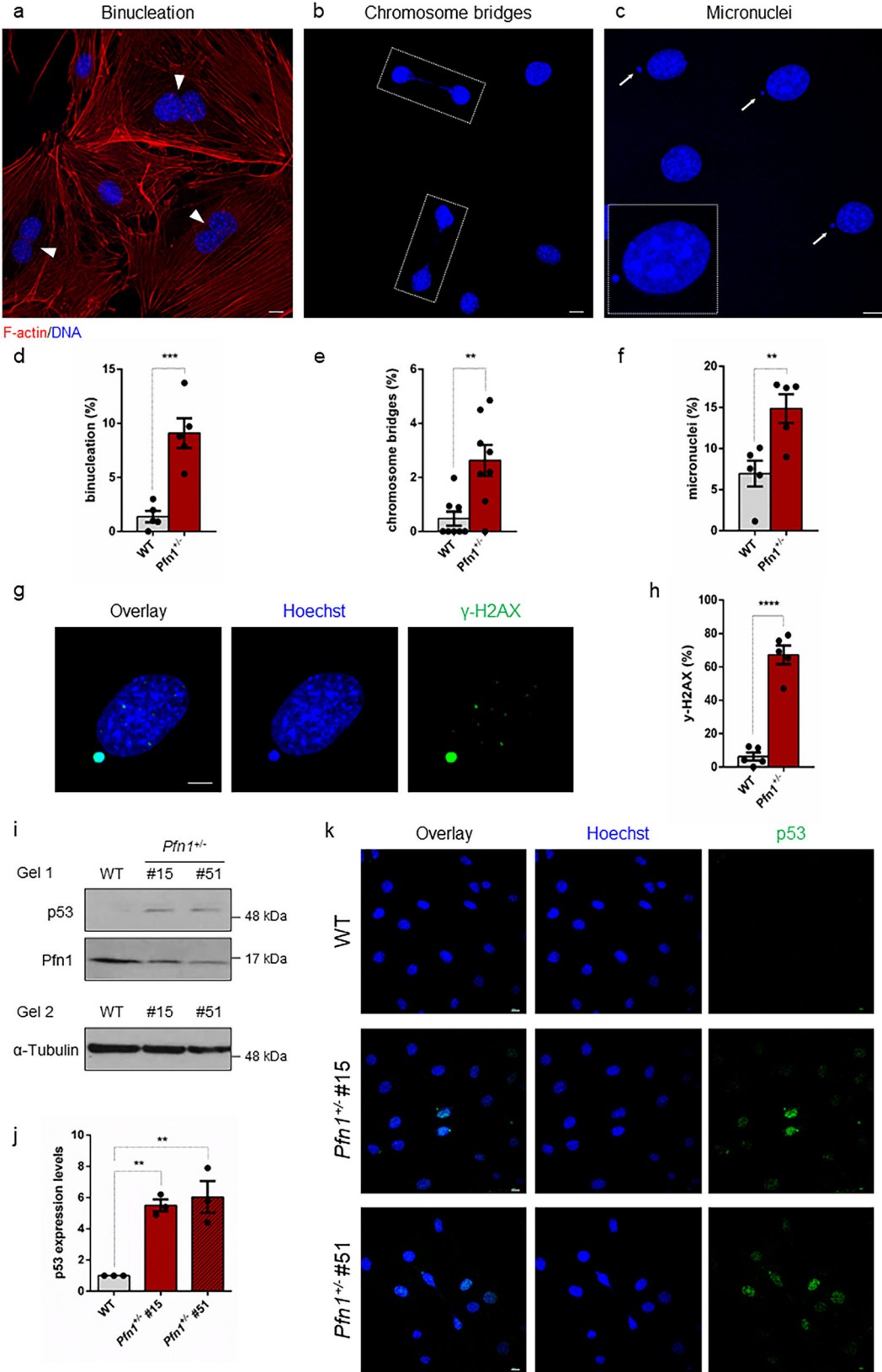

8/10 tumours with LOH of *PFN1*, it is more likely that loss of *PFN1* occurred before WGD rather than through multiple chromosomal losses following WGD, although we cannot fully rule out either possibility. Next, we sought to investigate when WGD events occurred during tumour evolution. To this aim, we integrated somatic point mutation and copy-number data to perform both relative and real-time timing analysis of WGD events (Methods section)[64,65]. The key idea is that somatic mutations acquired before WGD should be present in 2 chromosomal copies, whereas mutations occurring after WGD should only be present in one copy. By assuming a constant rate of mutation accumulation during tumour evolution, which allows

**Fig. 5 Profilin 1 deficiency is associated with nuclear abnormalities and DNA damage.** Representative confocal images of $Pfn1^{+/-}$ MC3T3 cells subjected to phalloidin (F-actin, red) and Hoechst 33342 (nucleus, blue) staining, showing nuclear abnormalities: **a** binucleated cells are indicated by arrowheads; **b** dashed lines indicate chromosome bridges connecting two cells; **c** micronuclei are indicated by white arrows; scale bars 10 μm. **d** Quantification (%) of binucleated cells; data are shown as mean ± s.e.m.; dots represent the value of each experiment ($n = 508$ WT and 479 KO cells; range of number of cells used for each individual experiment: 94-112 WT, 89-103 $Pfn1^{+/-}$); ***$P = 0.0004$. **e** Quantification (%) of cells with chromosome bridges; data are shown as mean ± s.e.m.; dots represent the value of each experiment ($n = 800$ WT and 771 KO cells; range of number of cells used for each individual experiment: 88-112 WT, 89-111 $Pfn1^{+/-}$); **$P = 0.0021$. **f** Quantification (%) of micronucleated cells; data are shown as mean ± s.e.m.; dots represent the value of each experiment ($n = 485$ WT and 496 KO cells; range of number of cells used for each individual experiment: 74-119 WT, 89-111 $Pfn1^{+/-}$); **$P = 0.0049$. **g** Confocal image of a micronucleated $Pfn1^{+/-}$ MC3T3 cell stained for DNA (Hoechst 33342, blue) and γ-H2AX (green); scale bar 5 μm. **h** The mean percentage of cells positive to γ-H2AX is shown as mean ± s.e.m.; dots represent the value of each experiment ($n = 169$ WT and 324 KO micronucleated cells; range of number of cells used for each individual experiment: 17-47 WT, 38-82 $Pfn1^{+/-}$); ****$P < 0.0001$. Data in **d**-**f**, **h** were analysed by one-tailed unpaired Student's t-tests; results from $Pfn1^{+/-}$ cell clones (#15 and #51) were grouped into one data set. **i** Western blotting analysis of WT and 2 different $Pfn1^{+/-}$ MC3T3 clones (#15 and #51) showing p53 accumulation. Due to similar molecular weight of ~50 kDa between p53 and α-Tubulin, the same amount of protein was loaded two times on the same gel, and α-Tubulin was used as loading control. **j** Quantification of p53 normalised to α-Tubulin; bars represent mean ± s.e.m.; dots represent the mean for each experiment ($n = 3$ from two independent protein extracts); **$P = 0.0041$, **$P = 0.0023$. Ordinary one-way ANOVA test performed. **k** Confocal images of p53 nuclear localisation in WT and $Pfn1^{+/-}$ clones (#15 and #51). DNA is stained with Hoechst 33342 (blue); scale bars 10 μm.

to perform real-time estimates for WGD events relative to the age at diagnosis[65], we found that WGD events occurred decades before diagnosis in the 4 patients (ranging from 20 to 48 years before diagnosis), and in the 2 cases with 2 WGD events, the estimated time between doublings was also in the order of decades (Fig. 7e and Supplementary Data 1).

Together, these data indicate that OS in PDB patients shows remarkably complex genome aberrations, including WGD events and chromosomal rearrangements.

## Discussion

Animal cells undergo dramatic morphological changes as they progress through the different stages of mitosis[3]. The actin cytoskeleton contributes to spindle morphogenesis and positioning[66-68] and thus, deregulation of actin dynamics could jeopardise chromosome segregation. The mitotic spherical shape is achieved through an increase in the cortical tension at prophase, triggered by RhoA-mediated phosphorylation of DIAPH1, which utilises Profilin 1 to polymerise actin at the cortex[52]. Therefore, we here hypothesised that Profilin 1 inactivation, which has been observed in diverse types of carcinomas[31-37] and sarcomas[26,69], might result in cytoskeletal defects that lead to CIN. We used both CRISPR/Cas9 and shRNA experiments to inactivate *PFN1* expression in RPE1 cells, which are widely used to study mitotic defects and chromosomal rearrangements, although in a p53-deficient background[12,70-72], which was not needed in our study. Our data show that Profilin 1 contributes to mitotic cell rounding, since Profilin 1-deficient cells struggle to round up, and the transition from G2 to mitosis requires several attempts, delaying cell division. The observed defects in mitotic cell rounding were accompanied by a remarkable reduction in cortical actin and, more importantly, chromosome mis-segregation events. Interestingly, frequency of mitotic defects was similar between *PFN1* heterozygous and homozygous KO RPE1 cells, indicating that the loss of one functional gene copy is sufficient to drive the errors, which is consistent with *PFN1* as a haploinsufficient gene. Thus, balanced levels of Profilin 1 must be ensured within the cells to allow for correct cell division. As proof of this, overexpression of wild type Profilin 1 still delayed mitotic progression, probably because of excessive actin nucleation that stiffens cell cortex. The loss of chromosome segregation fidelity induces CIN[2], a hallmark of many tumours and especially osteosarcomas[7,17,18,25]. Consistent with this, mitotic dysfunction and SCNAs were also detected in mesenchymal cells depleted for *Pfn1*, both in vitro (MC3T3) and ex vivo (MEFs and calvaria-derived cells). Of note, complete inactivation of *PFN1* was

achieved exclusively in the immortalised RPE1 cells and in other tumoral cell lines not described in the current study, while only heterozygous knock-out was obtained in the primary MC3T3 cells. This observation implies that complete Profilin 1 loss is tolerated only in transformed cells.

Homozygous *Pfn1* knock-in mice are not viable, while, curiously, heterozygous mice grow normally and display no overt tumour phenotype. Nonetheless, we noticed that the ratio of $Pfn1^{c.318\_321del/WT}$ to $Pfn1^{WT/WT}$ in the viable offspring was lower than expected when crossing wild type and heterozygous animals (observed 1:0.6 vs expected 1:1), suggesting that some $Pfn1^{c.318\_321del/WT}$ defective embryos could be selected during development. Furthermore, the evidence that haploinsufficiency of *Pfn1* did not drive tumorigenesis within 18 months of age is not surprising, mainly in animals with functional p53. Indeed, haploinsufficiency for most of the genes involved in the maintenance of chromosome stability (e.g. *Cenpe*, *Mad1*, *Mad2*, *Plk4*, *Bub1*) leads to long tumour latency and low penetrance, despite the high percentage of aneuploid cells[73-78], suggesting that there are factors (e.g. p53) that restrain the transforming potential of aneuploidy[78,79]. Here, we found that chronic Profilin 1 deficiency triggered the activation of the p53 pathway, and floating dead cells were observed in *PFN1*-KO cell cultures, likely as a result of p53-induced apoptosis. Nevertheless, our results demonstrate that most abnormal mitoses did result in viable daughter cells, implying that Profilin 1 inactivation causes mitotic errors even in a p53-proficient background.

In all cell models analysed, we pointed out occasional cytokinesis defects resulting in binucleated daughter cells. This result could be explained by insufficient accumulation of F-actin in the cleavage furrow of *PFN1*-KO RPE1 cells as a result of poor recruitment of Profilin 1 to the spindle midzone. This result is consistent with previous findings showing frequent binucleation in chondrocytes from mice conditionally depleted for *Pfn1* in cartilage[39]. While cytokinesis is essential for cell proliferation, an important fraction of cancers likely result from a cytokinesis failure[80,81]. Accordingly, WGD events and LOH at the *PFN1* locus were frequently detected in OS/PDB tumours. Our analysis revealed that *PFN1* loss likely occurred before WGD, which could result from cytokinesis failures induced by Profilin 1 reduction. In addition to WGD, we observed extensive SCNAs and massive genomic rearrangements in OS/PDB samples, including chromothripsis. Although we cannot causally link the presence of complex rearrangements and loss of *PFN1* in clinical samples given that *TP53* is also inactivated in these cases, our results are in line with the high frequency of chromosome bridges and

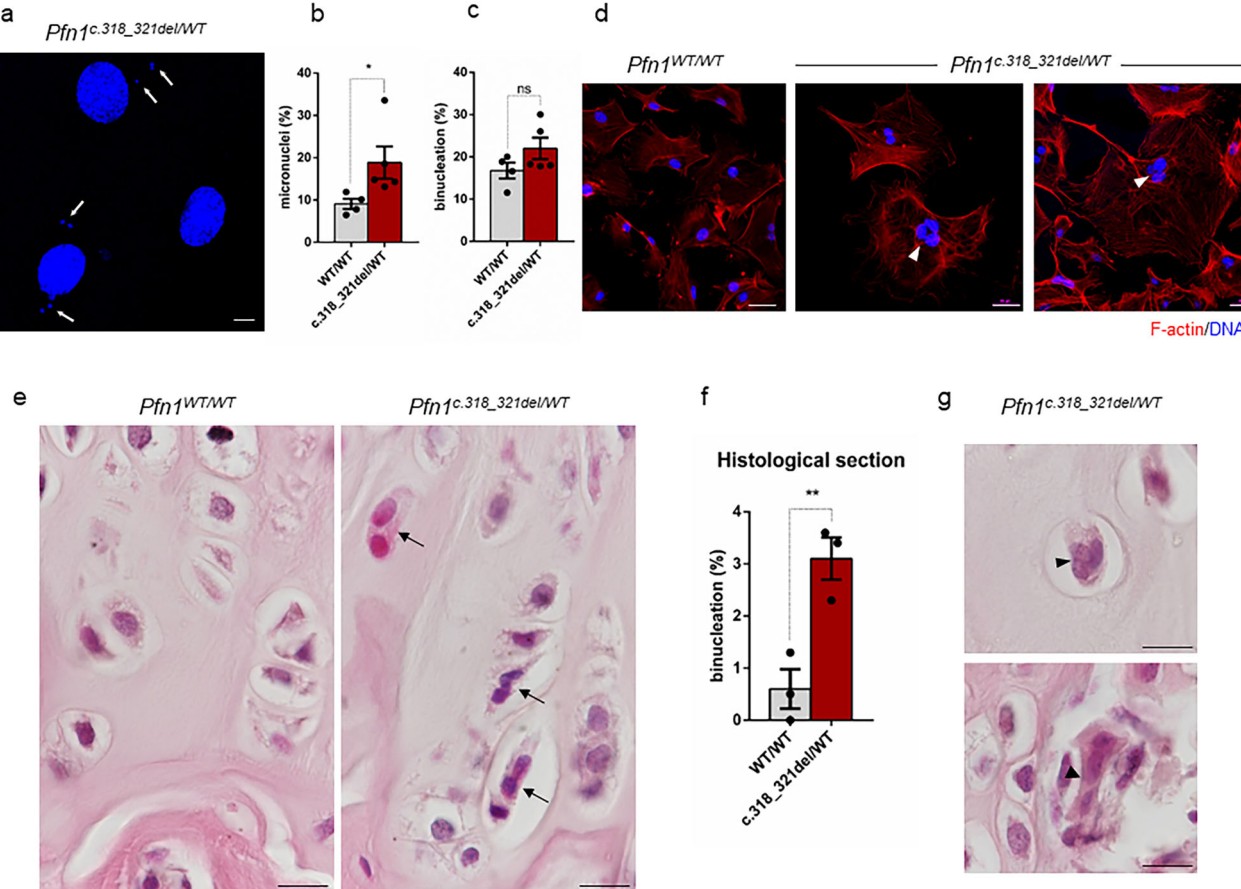

**Fig. 6 Profilin 1 insufficiency underlies cell division defects in vivo. a** Representative images of micronuclei (white arrows) in mesenchymal cells isolated from the calvarias of *Pfn1* knock-in (c.318_321del/WT) mice; scale bar 10 μm. **b** Quantification (%) of micronuclei in normal (WT/WT) and Profilin 1-deficient (c.318_321del/WT) mesenchymal cells; data are shown as mean ± s.e.m.; dots represent the value of each experiment ($n = 4$ biological replicates for WT samples, five biological replicates for mutant samples; scoring a total of 485 WT and 464 mutant cells; range of number of cells used for each individual experiment: 93-164 WT/WT, 67-107 c.318_321del/WT); *$P = 0.0312$. One-tailed unpaired Student's *t* tests performed. **c** Quantification (%) of binucleation in normal (WT/WT) and Profilin 1-deficient (c.318_321del/WT) mesenchymal cells; data are shown as mean ± s.e.m.; dots represent the value of each experiment ($n = 4$ biological replicates for WT samples, five biological replicates for mutant samples; scoring a total of 646 WT and 711 mutant cells; range of number of cells used for each individual experiment: 103-234 WT/WT, 102-169 c.318_321del/WT); ns $P = 0.0778$. One-tailed unpaired Student's *t* tests performed. **d** Images of primary *Pfn1^WT/WT^* and *Pfn1^c.318_321del/WT^* mesenchymal cells stained for F-actin (phalloidin, red) and nuclei (Hoechst 33342, blue); scale bars 50 μm. Arrowheads point towards trinucleated cells. Note that Profilin 1 insufficiency results in large cells with a loose actin cytoskeleton. **e** Representative haematoxylin/eosin-stained sections of the femoral epiphyseal growth plate of 4-month-old control (*Pfn1^WT/WT^*) and knock-in (*Pfn1^c.318_321del/WT^*) mice; scale bars 100 μm. Arrows point towards binucleated cells. **f** Quantification (%) of binucleated cells in the femoral growth plate of control (*Pfn1^WT/WT^*) and knock-in (*Pfn1^c.318_321del/WT^*) mice; data are shown as mean ± s.e.m.; dots represent the value of each experiment ($n = 3$ biological replicates; scoring ≥ 61 cells per replicate; **$P = 0.0054$. One-tailed unpaired Student's *t* tests performed). **g** Haematoxylin/eosin-stained sections showing trinucleated cells (arrowheads) in 4-month-old *Pfn1^c.318_321del/WT^* mice; scale bar 100 μm.

micronuclei found in Profilin 1-deficient cells, both reported to contribute to genome instability and chromothripsis[72,82,83].

In conclusion, we propose that Profilin 1 inactivation results in defective cytoskeleton organisation, thereby affecting mitotic entry, progression, and exit. Our work mechanistically links Profilin 1-dependent mitotic defects to CIN in epithelial and mesenchymal cell lines, and mouse experiments. By uncovering mitotic defects without inactivating p53, we here also highlight *PFN1* knock-out as a tool to study aneuploidy and complex genome rearrangements in a p53-proficient background.

## Methods
**Cell culture and treatment**. All cell lines were maintained in a humidified incubator at 37 °C with 5% $CO_2$. hTERT-RPE1 (referred to as RPE1) and derivatives were grown in DMEM/F12 (Thermo Fisher, 11320033) supplemented with 10% FBS and 0.01 mg/ml Hygromycin B (Thermo Fisher, 10687010); RPE1-derived shRNA clones were cultured in the presence of 0.4 mg/ml neomycin. To synchronise RPE1 cells, cells were treated with 5 mM thymidine (Sigma-Aldrich,

T1895) for 16 h, released for 8 h, and then treated again with thymidine for 16 h. After 2 washes with 1x PBS (Thermo Fisher, 14190169), cells were cultured in standard DMEM/F12 medium until entering mitosis, fixed and subjected to immunofluorescence experiments. Based on our experience, WT and CRISPR-WT RPE1 cells accumulate at M phase after a 7-h release into normal medium, *PFN1^+/−^* RPE1 cells enter M phase after 8/9-h release, whereas the majority of *PFN1^−/−^* RPE1 cells peak at mitosis after 12-14 h of release. In live cell imaging experiments, cells were recorded after 2 h from the second release in thymidine-free medium. In Supplementary Fig. 3b, RPE1 cells were synchronised in G0 by serum starvation to discriminate between tetraploid peaks and cells with duplicated DNA (G2/M): cells were plated at 50-70% confluency, washed three times with 1x PBS, and starved in DMEM with 0% FBS for 48 h. To analyse mitotic rounding on collagen, RPE1 cells were grown on glass coverslips coated with 50 μg/ml rat-tail collagen I (Gibco, A1048301). Then, the cells were synchronised in mitosis using a thymidine-nocodazole block: cells were treated with 2 mM thymidine for 30 h, released for 6 h (WT) or 10 h (*PFN1*-KO), and then treated with 50 ng/ml nocodazole (Sigma-Aldrich, M1404) for 4 h. MC3T3 and derivatives were maintained in Minimum Essential Medium α (MEM-α) without ascorbic acid (Thermo Fisher, A1049001) with 10% FBS, 1% penicillin–streptomycin, and 0.5% gentamycin. To synchronise MC3T3 cells in early S phase, cells were cultured for 30 h in 2 mM thymidine, washed 2 times in 1x PBS and released into the standard MEM-α

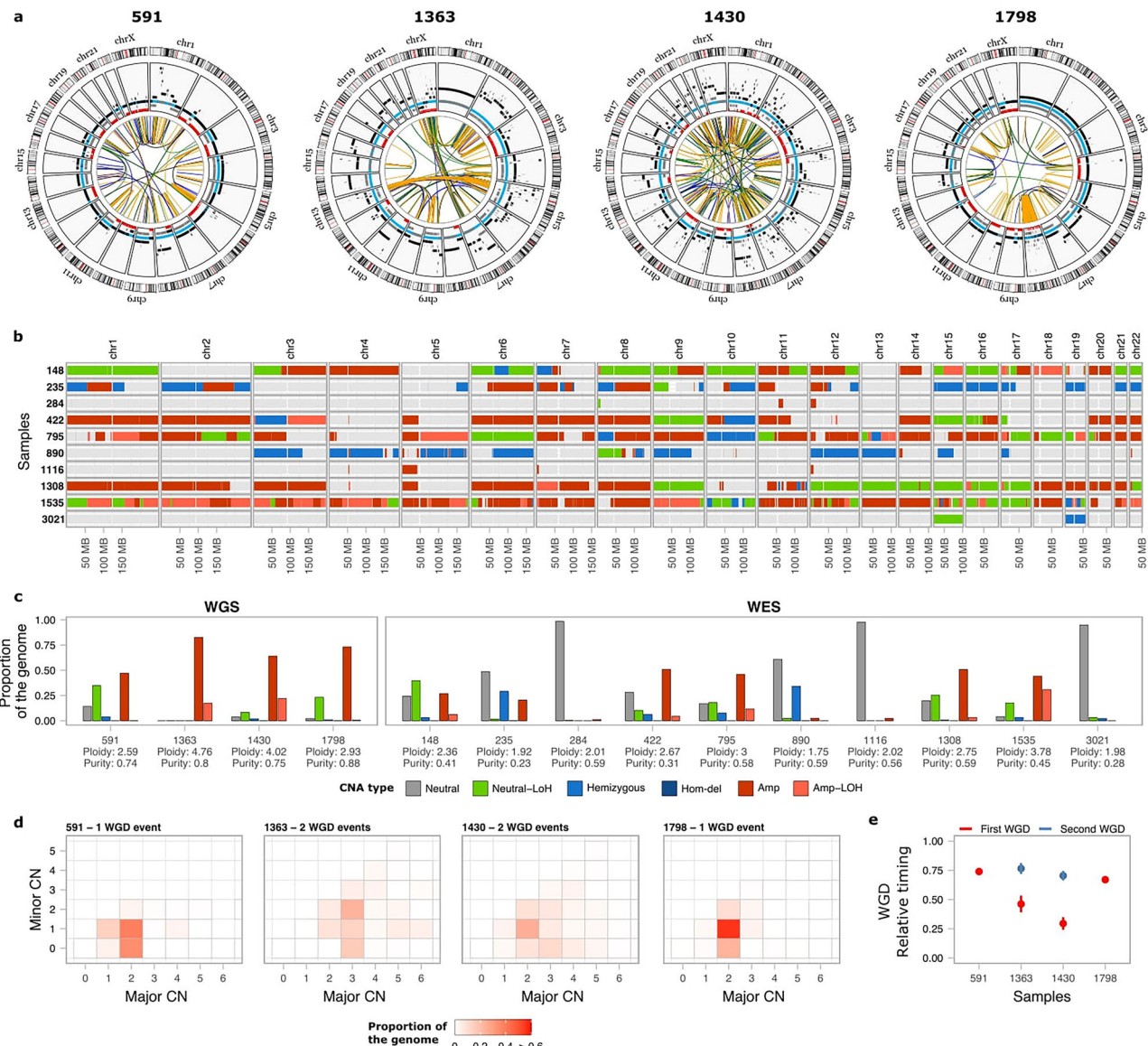

**Fig. 7 Somatic genomic landscape of 14 OS/PDB patients. a** Circos plots for 4 OS/PDB patients analysed using whole-genome sequencing. The inner circle shows somatic SVs coloured according to the type of SV (duplication-like, deletion-like, head-to-head and tail-to-tail inversions are shown in blue, orange, black and green, respectively). The second circle shows SCNAs, with black representing the total copy number, grey the minor copy number, and red the minor copy number when equal to 0. Blue segments mark the expected normal diploid state (CN = 2) as a reference for visualisation. **b** SCNA calls for the 10 WES OS/PDB patients. The colour of the bars represents the different types of SCNAs. **c** Proportion of the genome affected by each type of SCNA. **d** Heatmap of the proportion of the genome representing the different combinations of the obtained major and minor copy numbers in the WGS samples used to estimate the number of WGD events. **e** Relative timing analysis of the WGD events for the WGS samples. The colour of the dots represents the first (red) and second (blue) WGS events for each sample. Lines represent the 95% confidence intervals after bootstrapping 1000 times. Neutral: total copy number = 2 and minor copy number = 1; Neutral-LoH: total copy number = 2 and minor copy number = 0; Hemizygous: total copy number = 1; Hom-del: homozygous deletion; Amp-LoH: total copy number > 2 and minor copy number = 0.

medium; after 2 h, cells were subjected to live cell imaging. HEK293T (293T) were cultured in DMEM High-Glucose (Thermo Fisher, 11965084) with 10% FBS and 1% penicillin–streptomycin. Human dermal fibroblasts were previously isolated from a skin biopsy of a healthy donor and were drawn from stock at the biobank available at our Institute (https://www.igb.cnr.it/index.php/pagets-disease-of-bone-biobank/). Cells were cultured in DMEM High-Glucose (Thermo Fisher, 41965062) with 10% FBS and 1% penicillin–streptomycin. HK-2 cells were grown in DMEM/F12 supplemented with 5% FBS, 5% Insulin-Transferrin-Selenium-Sodium Pyruvate (Thermo Fisher, 51300044), 1% penicillin–streptomycin, and 1% L-Glutamine. Human peripheral blood mononuclear cells (PBMCs) were isolated from a healthy donor by density-gradient centrifugation using Ficoll-Histopaque solution (Sigma-Aldrich, H8889), under the approval of the Ethics Committee for Biomedical Activities "Carlo Romano," International Office for Bioethics Research, University of Naples "Federico II". The donor signed an informed consent before participating to the study. All cell lines were tested

for mycoplasma contamination (PCR Mycoplasma Detection Kit, Applied Biological Materials, G238) and were found to be negative. Cell lines were not authenticated.

**Generation of a _Pfn1_ knock-in mouse model.** The 129/Sv genomic DNA was used as a template to amplify the _Pfn1_ locus for the homologous recombination. The _Pfn1_ c.318_321delTGCC mutation was introduced in the targeting vector by PCR-mediated mutagenesis (QuikChange Lightning, Agilent). Targeting of the construct was done into the E14Tg2a embryonal stem cell line, and targeted ES cell clones were identified by Southern blot analysis using a 3′ probe and an internal Neo probe on EcoRI-cut genomic DNA, and a 5′ probe on BamHI-cut genomic DNA. One _Pfn1_<sup>c.318_321del/WT</sup> embryonic stem cell clone was injected into C57BL6 blastocysts to establish a mutant mouse colony. These mice were maintained on a mixed genetic background. Geno-typing was performed with allele-specific primers: forward 5′-AACTCCAGCT CCACAGTACATAAG-3′; reverse 5′-TTATGCAGCCTTGACACTGAGGAC-3′.

Heterozygous mutant crosses did not yield any homozygous mutant animals at birth ($n > 200$ live births) or as embryos ($n = 44$). To establish the ratio of $Pfn1^{c.318\_321del/WT}$ to $Pfn1^{WT/WT}$ mice in the viable offspring, 193 animals were counted from 25 matings. All procedures were performed in accordance with the guidelines approved by the Italian Ministry of Health (Protocol #125-2021-PR) and were approved by the Institute of Genetics and Biophysics (IGB) Institutional Animal Care and Use Committee (IACUC). All mice were housed in a pathogen-free barrier environment.

**Isolation of mouse embryonic fibroblasts.** $Pfn1^{c.318\_321del/WT}$ heterozygous mice were crossed; pregnant females were sacrificed at 14.5 days after the appearance of the copulation plug by cervical dislocation. Uterine horns were dissected out and placed in tubes containing PBS + 1% penicillin–streptomycin. Embryos were separated from their placenta and surrounding membranes. Red organs (heart and liver) and tail (for genotyping) were removed. Embryos were finely minced using razor blades and suspended in a tube containing 3 ml 0.05% trypsin and incubated at 37 °C for 15 min in a shaking water bath. Two volumes of media (DMEM supplemented with 10% FBS, 1% penicillin–streptomycin) were added to inactivate the trypsin, and remaining pieces were removed by letting them to settle down to the bottom of the tube within 5 min. The supernatant was carefully taken off to be subjected to low-speed centrifugation. Cell pellet was resuspended in warm medium and plated in 25 cm² flasks.

**Isolation and culture of neonatal mouse calvaria-derived cells.** Primary murine mesenchymal stromal cells (mMSCs) were isolated from neonatal mouse calvarias. Mice pups (P2/3) were euthanized by inhalation of CO₂, heads from $Pfn1^{WT/WT}$ and $Pfn1^{c.318\_321del/WT}$ were individually placed in a petri dish with 1x PBS, calvaria were cleaned from adherent soft tissues and subjected to sequential digestion in 1x PBS containing 1 mg/ml collagenase type IV (Sigma-Aldrich, C5138), 0.025% trypsin (Biowest, L0909) and 1% penicillin–streptomycin, at 37 °C in shaking water bath. The cells from digestions 2–4 were collected, pooled and cultured in MEM-α without ascorbic acid with 10% FBS and 1% penicillin–streptomycin; cells were maintained in culture for no more than 4 passages.

**Targeted knock-out of PFN1 using CRISPR-Cas9.** PFN1-KO RPE1 cell clones were generated using the CRISPR guide RNA (gRNA) 5′- CACCGTTCGTAC-CAAGAGCACCGGT-3′. Annealed gRNA oligonucleotides were inserted into the pSpCas9n(BB)-2A-GFP plasmid (Addgene, PX458) and the construct was transfected into RPE1 cells using Lipofectamine LTX with PlusReagent (Invitrogen) transfection reagent. GFP-positive single cells were sorted by FACS into 96-well plates. Candidate single-clone colonies were verified by Sanger sequencing and western blotting. Off-target mutations in genes predicted by Optimized CRISPR design tool at http://crispr.mit.edu/ (GRAMD4, INTS6) were excluded through Sanger sequencing.

**PFN1 silencing.** To transiently knockdown Profilin 1 expression in RPE1 cells, a custom pLKO.1-puro-CMV-TurboGFP was purchased from Sigma-Aldrich, expressing a shRNA previously shown as efficiently targeting the CDS of PFN1 (5′-CCGGCGGTGGTTTGATCAACAAGAACTCGAGTTCTTGTTGATCAA ACCACCGTTTTT-3′)[84]. As a negative control, the same pLKO-based plasmid expressing a non-target shRNA was purchased from Sigma-Aldrich and used. Stable RPE1 clones were generated through Nucleofection using the Amaxa Cell Line Nucleofector Kit V (VCA-1003), following the X-001 protocol provided by Amaxa Nucleofector (Lonza). After 24 h, clones stably expressing either the non-target or the PFN1 shRNA were selected with neomycin (0.5 mg/ml) every 60 h for 10 days. Single cell clones were picked and expanded; colonies from a single cell were then maintained with 0.4 mg/ml neomycin. Silencing was verified through western blotting. Three non-target (shControl) and five shPFN1 RPE1 clones were subjected to the experiments.

To conditionally silence PFN1, RPE1 cells expressing the doxycycline-inducible shRNA were cultured in medium containing 2.5 µg/ml doxycycline (Sigma-Aldrich, D9891), refreshing media every day for 7 days or for the time indicated in the figures.

**Lentiviral infection.** RPE1 cells stably expressing PFN1^WT, the actin-binding mutant PFN1^R89E, the microtubule-binding mutant PFN1^G118V, and H2B-mCherry/EGFP-Tubulin were generated by lentiviral transduction. The lentiviral plasmid expressing wild type Profilin 1 fused to GFP was purchased from Origene (pLenti-PFN1-C-mGFP, # RC202338L2). Site-directed mutagenesis introduced the R89E and the G118V mutations into pLenti-PFN1-C-mGFP, using the following primers:

(R89E) 5′-CAGGATGGGGAATTTAGCATGGATCTTGAAACCAAGA GCACCGG-3′;

(G118V) 5′-GCTGATGGGCAAAGAAGTTGTCCACGGTGGTTTG-3′.

Sanger sequencing was used to confirm the presence of mutations, and western blotting proved overexpression of the distinct forms Profilin 1 (Supplementary Fig. 12). Lentiviral plasmids expressing H2B-mCherry or EGFP-Tubulin were purchased by Addgene (CSII-prEF1a-mCherry-3xNLS, #125262; L304-EGFP-Tubulin-WT, #64060). pCMV-VSV-G and pCMV-ΔR8.2 co-packaging plasmids were a gift from Dr Annalisa Fico. Lentiviruses expressing the respective genes were

generated by cotransfection of 293T cells with pCMV-VSV-G and pCMV-ΔR8.2, using Lipofectamine 2000 (Invitrogen). The individual lentiviral supernatants were collected after 24 and 48 h, and then used for infection of target RPE1 cells. Infected cells ($5 \times 10^4$) exhibiting GFP fluorescence (for PFN1 overexpressing cells) or both mCherry and GFP fluorescence (for H2B-Tubulin expressing cells) were selected through FACS Aria (Becton Dickinson) and further expanded.

Doxycycline inducible shRNA construct for PFN1 was generated by cloning the sequence 5′-GCATGGATCTTCGTACCAAGA-3′ in the Tet-pLKO-neo plasmid (Addgene #21916). For virus production, lentiviral plasmids were transfected into 293T cells with lentivirus packaging plasmids, and supernatant medium containing virus was collected at 36 h after transfection. RPE1 cells were infected with lentivirus and medium was changed 6 h after infection. Infected cells were selected by neomycin selection after 7 days from infection.

**Time-lapse microscopy.** For live cell imaging, RPE1, MC3T3, and MEF cells were cultured as described above. For phase-contrast time-lapse analysis, RPE1 and MC3T3 were synchronised at the G1/S boundary by the thymidine block method, as described above. At 2 h from the release in thymidine-free medium, the cells regained the cell cycle progression in a microscope stage incubator at 37 °C in a humidified atmosphere of 5% CO₂ throughout the experiment. MEF cells were analysed as asynchronous population due to the accumulation of senescent cells observed in thymidine-treated MEF cultures. Cells were cultured in six-well plates ($4 \times 10^4$ RPE1; $6 \times 10^4$ MC3T3; $1.2 \times 10^5$ MEFs). Time-lapse images were acquired with an inverted Zeiss Axio Observer Z1 widefield microscope equipped with an AxioCam MRm grayscale CCD camera and controlled by ZEN pro software (Zeiss). Phase-contrast images were captured as Z-stacks (10 planes, 3 µm interval) every 2 min for 15-20 h, using a x20 objective. The resulting images were processed and analysed using ZEN pro (Zeiss) software. For fluorescence live cell imaging, $6 \times 10^4$ asynchronous H2B-mCherry/EGFP-Tubulin RPE1 cells were plated on a 6-well plate 20 h before the acquisition, and then placed on the stage of a THUNDER Imager Live Cell (Leica) equipped with an environmental chamber which provided an adequate temperature, humidity, and CO₂ control. Time-lapse images were captured as Z-stacks (12 planes, 2.4 µm interval) at 3 positions every 2 min for ~20 h, using a x20 objective and 535/590 excitation filters.

**Protein isolation and western blotting.** Total cell lysates (RPE1, MC3T3) were isolated using RIPA buffer supplemented with 1:100 protease (Applied Biological Materials, G135). Mouse tissue disruption and homogenisation was performed in RIPA buffer supplemented with 1:100 protease using Tissue Lyser LT (Qiagen), for 5 min at 50 Hz. Protein concentrations were determined using Bradford reagent (Bio-Rad). Lysates were boiled at 95 °C for 5 min and separated by SDS–PAGE electrophoresis using 8-16% Tris-Glycine gels (Invitrogen) at 225 V for 30 min. Samples were then transferred onto nitrocellulose membranes using an iBlot transfer system (Invitrogen) for 5 min, and blocked using 5% w/v nonfat dry milk dissolved in TBST (1X TBS, 0.05% Tween-20) for 1 h at room temperature. Membranes were then incubated with the following primary antibodies: rabbit anti-Profilin 1 (Novus Biologicals, NB200-162, 1:10000 for human lysates; Invitrogen, PA5-17444, 1:5000 for murine lysates); mouse anti-p53 (Cell Signaling, 2524, 1:1000); mouse anti-α-Tubulin (Sigma, T6074; 1:15000); mouse anti-GAPDH (SC-32233; 1:1000). Membranes were then incubated with secondary antibodies conjugated with HRP for 1 h at room temperature. The bands were visualised using enhanced chemiluminescence detection reagents (Advansta) and autoradiographic films (Aurogene). Equal loading was confirmed by using antibody against α-Tubulin or GAPDH. The intensity of the western blot signals was determined by densitometry analysis using the ImageJ software and normalised to the density value of the loading control (α-Tubulin).

**Indirect immunofluorescence.** Cells were grown on coverslips, washed in 1x PBS and fixed in 4% paraformaldehyde for 15 min at room temperature. Coverslips were blocked in 4% FBS in 1x PBS + 0.1% Triton X-100 prior to incubation with primary antibodies diluted in blocking buffer: rabbit anti-γ-H2AX (Cell Signaling, 2577, 1:400); mouse anti-p53 (Cell Signaling, 2524, 1:2000); mouse anti-α-Tubulin (Sigma, T6074; 1:500); rabbit anti-Profilin 1 (Novus Biologicals, NB200-162, 1:100); and mouse anti-Aurora B (BD Transduction Laboratories, 611082, 1:100) overnight at 4 °C. Profilin 1 localisation at the spindle midzone was confirmed with a different antibody: rabbit anti-Profilin 1 (Invitrogen, PA5-17444, 1:50). An aliquot of the cells subjected to FACS analysis in Supplementary Fig. 3b was subjected to indirect immunofluorescence with the mitotic marker phospho-Histone H3 (Cell Signaling, 53348; 1:1600) to exclude that cells escaped serum starvation. Samples were washed three times in 1x PBS and primary antibodies were detected using species-specific fluorescent secondary antibodies (Life Technologies). Samples were washed 3 more times in 1x PBS before DNA detection with 1 µg/ml Hoechst 33342 (Invitrogen, H3570) for 15 min at room temperature. Cells were stained with 5 µM rhodamine phalloidin (Sigma, P1951) for 15 min at room temperature to visualise F-actin. Coverslips were washed 2 times in 1X PBS and then in distilled water and mounted with Mowiol. The cells were then examined using a Nikon's A1R confocal laser microscope. NIS-Elements Advanced Research Analysis software was used to quantify phalloidin fluorescence intensity.

**Flow cytometry analysis of the DNA content**. After serum starvation, the RPE1 cells were harvested and washed with cold PBS. Cells were fixed in ice-cold 100% methanol and incubated overnight at −20 °C. For FACS analysis, fixed cells were resuspended in PBS containing 0.2 mg/ml RNase, and incubated for 1 h at 37 °C. Then, cells were stained with 50 mg/ml propidium iodide for 30 min. DNA content was analysed by flow cytometry (Becton Dickinson FACSAria). Human PBMCs were used as a diploid internal standard to accurately identify the G0/G1 diploid peak position.

**Haematoxylin and eosin staining**. Haematoxylin and eosin staining was conducted on paraffin-embedded tissue. All sections used for histological analysis were 5-μm thick. Sections were deparaffinised using xylene for 30 min and rehydrated through a graded series of alcohol (100, 90, 80 and 70% alcohol, each for 5 min). The slices were then incubated with haematoxylin solution (Sigma Aldrich) for 1 min and, after rinsing in distilled water, with eosin for 3 min. Afterwards, the sections were dehydrated in ethyl alcohol solution of gradient concentrations, embedded in the mounting medium, covered with a coverslip and analysed by transmission light microscopy (Nikon Intensilight C-HGFI).

**Scoring of cellular abnormalities**. Mitotic defects (misaligned or lagging chromosomes, multipolar metaphases, anaphase bridges) were counted manually from Hoechst-stained CRISPR RPE1 cells visualised at Nikon's A1R confocal laser microscope. Experiments were repeated three times and on average 100 cells were counted for each experiment. Anaphase bridges and lagging chromosomes in shRNA RPE1 were counted in two independent experiment sets and ≥ 30 mitoses were analysed in each experiment. Micronucleated cells, binucleated cells, and cells with chromosome bridges were counted manually from Hoechst-stained MC3T3 visualised at Nikon's A1R confocal laser microscope. Criteria for micronuclei scoring were as follows: (i) diameter of micronuclei less than one-third of the nucleus; (ii) intensity of Hoechst-stained micronuclei similar to that of the main nucleus; (iii) micronuclear boundary distinguishable from the nuclear. Experiments were repeated at least five times and at least 100 cells were counted for each experiment. Due to similar scoring, results from the two $Pfn1^{+/-}$ cell clones (#15 and #51) were grouped into one data set. Micronucleation and binucleation in primary mesenchymal cells derived from WT and knock-in mice were assessed manually from Hoechst-positive cells. At least four biological replicates per genotype were analysed and about 100–200 cells were counted for each experiment. Binucleated cells within the femoral sections were scored using three biological replicates per genotype; at least five slices per sample were analysed and on average 50 cells per field were counted. Aberrant mitotic rounding was quantified by visual inspection of partially rounded or flat mitotic cells at metaphase. The experiment was carried out two times, scoring a total of 206 WT, 260 $PFN1^{+/-}$, and 139 $PFN1^{-/-}$ RPE1 cells on uncoated coverslips, and a total of 244 WT, 238 $PFN1^{+/-}$, and 177 $PFN1^{-/-}$ RPE1 cells on collagen-coated coverslips.

**Genomic DNA extraction**. Genomic DNA from OS/PDB tissues and matched healthy bone marrow samples was isolated using the High Pure PCR Template Preparation Kit following the manufacturer's instructions (Roche Life Science). Genomic DNA from cell lines (MC3T3 and RPE1) was extracted using the Proteinase K Method, incubating cells in the PK buffer (150 mM NaCl; 2 mM EDTA; 1% SDS; 20 mM Tris–HCl pH 8; 0.5 mg/mL Proteinase K) for 1 hour at 55 °C, and precipitating DNA with saturated NaCl and 100% EtOH. For WES samples, target enrichment was performed using the Agilent SureSelect Human All Exon V6 design (Agilent, Santa Clara, CA, USA).

**Whole-genome and whole-exome sequencing of OS from PDB patients**. Whole-genome and exome libraries were sequenced on the HiSeq 4000 platform (Illumina) in 151 bp paired-end mode. Raw sequencing reads were mapped to the GRCh38 build of the human reference genome using BWA-MEM version 0.7.17-r1188 [ref. [85]]. Aligned reads in BAM format were processed following the Genome Analysis Toolkit (GATK, version 4.1.8.0) Best Practices workflow to remove duplicates and recalibrate base quality scores[86]. Somatic point mutations and indels were detected using Mutect2[87] (GATK, version 4.1.8.0), MuSE[88] (version 1.0rc) and Strelka2 [ref. [89]] (version 2.9.2) using the matching germline data as control. Each algorithm was run independently on each tumour-normal pair, and the calls were integrated using the Python library mergevcf (https://github.com/ljdursi/mergevcf). Only mutations detected by at least 2/3 algorithms were considered for further analysis. For the WGS data, somatic copy-number aberrations, ploidy and purity values were detected using ascatNGS[90]. Structural variants were detected using Delly[91] (version 0.8.3), LUMPY[92] (version 9–120.2.13), Manta[93] (version 1.6.0), and SvABA[94] (version 1.1.3). Each tool was run independently on each tumour-normal pair. The calls generated by each algorithm were merged using mergevcf, and only structural variants detected by at least two algorithms were considered for further analysis. For WES samples, somatic copy-number aberrations were detected by integrating the output of GATK (version 4.1.8.0) and FreeBayes[95] (version 1.3) using PureCN (https://github.com/lima1/PureCN). Briefly, the GATK4 Somatic CNV workflow (GATK, version 4.1.8.0) was utilised for the normalisation of read counts and genome segmentation (using 2 unrelated normal samples as controls). FreeBayes was used to obtain B-allele frequency values for gnomAD (version 3.1) variant sites with population allele frequencies >0.1%. Finally, PureCN was used to integrate the output of GATK and FreeBayes to estimate the allele-specific consensus copy-number profile, purity and ploidy for each tumour sample.

For both WES and WGS samples, we considered that a tumour underwent one WGD if the copy-number of the major allele (i.e., the most amplified allele) was equal or greater than 2 in ≥50 % of the genome. Mutational states were integrated with the WGD analysis to infer both relative and real-time timing of WGD events[64,96]. The basic idea is that mutations occurring before a genome doubling would be present in multiple copies, whereas mutations occurring after WGD would be likely present in just one copy. Thus, the ratio of early versus late mutations informs about the timing of WGD during tumour evolution. Chromothripsis was detected using ShatterSeek[15].

**Whole-genome sequencing of $Pfn1^{+/-}$ MC3T3 clones**. Low-pass whole-genome sequencing (~7x) of the different $Pfn1$ knock-out clones and WT MC3T3 cells was performed on a HiSeq 4000 (Illumina) to obtain 100 bp paired-end reads. Raw sequencing reads were mapped to the GRCm38 build of the mouse reference genome using BWA-MEM version 0.7.17-r1188. Aligned reads in BAM format were processed following the Genome Analysis Toolkit (GATK, version 4.1.8.0) Best Practices workflow. Since the ploidy of the WT clone is ~4n, we also downloaded whole-genome data from the C57BL_6NJ line (ftp://ftp-mouse.sanger.ac.uk/current_bams/C57BL_6NJ.bam) to be used as a reference sample for copy-number analysis rather than the WT clone. The C57BL_6NJ data were downsampled to achieve a similar depth of coverage as the MC3T3 clones. Copy-number analysis was performed using CNVkit[97] (version 0.9.9). CNVkit was run using a window size of 50 Kbp and a ploidy of 4. Finally, chromosome-level copy number values were computed as the modal copy number for each chromosome normalised by segment size.

**Statistics and reproducibility**. All statistical analyses were performed using GraphPad Prism 6.0. All results are presented in graphs as the mean ± standard error of the mean (s.e.m.). Data were obtained from at least three independent experiments (unless otherwise noted in the figure legends), and have been overlaid with dot plots showing the values of each experiment. Each exact $n$ value is indicated in the corresponding figure legend. Western blot data presented are representative of at least three independent experiments that yielded similar results, unless otherwise noted in the figure legends. Immunofluorescence staining experiments were independently repeated at least three times; all the confocal images shown are representative of a minimum of 15 images. Student's $t$ test was used for comparing the statistical significance between 2 groups and all tests were determined using unpaired one-sided tests. Comparisons between multiple groups were assessed by one-way analysis of variance (ANOVA) with Dunnet's multiple comparisons test or two-way ANOVA with Bonferroni's or Sidak's multiple comparisons test. Differences between $PFN1^{+/-}$ and $PFN1^{-/-}$ RPE1 cells were statistically significant only when specified in the graphs. Specific $P$ values are labelled in the figure legends, where significant values are $P < 0.05$. No data were excluded from the analyses presented in this study.

**Reporting summary**. Further information on research design is available in the Nature Portfolio Reporting Summary linked to this article.

## Data availability

All relevant data supporting the key findings of this study are available within the article. Uncropped and unedited blot are shown in Supplementary Fig. 13 and 14. Source data are shown in Supplementary Data 2. Patients gave us permission to collect data in our Institute under the control of the Principal Investigator [F.G.] but not to release them in an unprotected database. Therefore, due to lack of approval, data related to the whole-exome and whole-genome sequencing at the individual level cannot be made public; however, they can be obtained from the corresponding author [F.G.] upon reasonable request. Whole-genome sequencing data of MC3T3 clones were deposited in the European Nucleotide Archive (ENA; https://www.ebi.ac.uk/ena) under the study project PRJEB58148, and individual accession numbers are indicated in Supplementary Data 4.

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

## Acknowledgements

Authors acknowledges members of the *Bone Diseases and Tumors* Laboratory at IGB-CNR for constructive feedbacks provided during manuscript revision. We also thank Dr Annalisa Fico (IGB-CNR) for her help and advice in the lentiviral infection of RPE1 cells. Authors are also grateful to Alessandro Sacco and Dr. Giuseppe Viglietto (Department of Experimental and Clinical Medicine, University Magna Græcia of Catanzaro, Italy) for their support during fluorescence time-lapse experiments. We sincerely acknowledge Dr. Silvia Taglietti (Department of Experimental Oncology at European Institute of Oncology, Milan, Italy) for her contribution in generating the RPE1 cell model expressing inducible shPFN1. We are grateful to members of the Integrated Microscopy, Mouse Modelling, and FACS Facilities of IGB-CNR. We thank the Euro-BioImaging Infrastructure at Institute for Experimental Endocrinology and Oncology (CNR), Naples, Italy for help with microscopy experiments. F.S.d.C. was supported by Fondazione Umberto Veronesi. F.S.d.C acknowledges funding from the European Calcified Tissue Society (ECTS) and from the Italian Society for Osteoporosis, Mineral Metabolism, and Skeletal Diseases (SIOMMMS). F.M. and I.C.-C. acknowledge funding from EMBL. The research leading to these results has received funding from AIRC under IG 2020 - ID. 25110 project – P.I. F.G.

## Author contributions

F.S.d.C. and F.G. conceived the study and wrote the paper with input from F.M. and I.C.-C. F.S.d.C., F.G., F.M. and I.C.-C. drafted the figures and tables. F.S.d.C. carried out in vitro and ex vivo experiments, and performed statistical analysis. S.R. performed histological analysis on mouse sections. F.S.d.C., S.R. and F.G. analysed the results. R.G. helped in interpretation of data about microscopy chromosome images. F.M. and I.C.-C. performed analyses of sequencing data. M.M., A.C.D.L. and F.B. helped in time-lapse imaging and provided microscopy infrastructures. L.G and S.S. designed and generated the inducible shPFN1-expressing RPE1 cells. L.P. and K.S. obtained clinical specimens and provided clinical information. I.C.-C. contributed to discussion. F.G. supervised the study and data analysis. All authors read and approved the final version of the manuscript.

## Competing interests

The authors declare no competing interests.
