## [Peer Review File · Communications Biology]

Reviewers' comments:

Reviewer #1 (Remarks to the Author):

This is an elegant study which performs a detailed mitotic characterization of in vitro and in vivo phenotypes of Pfn1 heterozygous and homozygous knockout cells demonstrating that loss of Pfn1 induces anaphase bridges, multipolar spindles, defects in cytokinesis and genome arrangements. While Pfn1's requirement for cleavage furrow formation and cytokinesis has been previously shown by another group (Bottcher et al. EMBO 2009), chromosomal changes upon loss of Pfn1 as demonstrated herein is novel. The paper is well-written and the conclusions are well-supported by experimental data. I have a few suggestions that would further improve the quality of the paper:

- 1) I would generally recommend WT-Pfn1-associated data next to KO phenotype data for comparison in all figures (the authors have done so in some but not for all the figures).
- 2) Interestingly, some of the metrics (including CIN) seem to be comparable between Pfn1 heterozygous and homozygous knockout cells – yet there is a wide range of Pfn1 expression between cell types and importantly mice with heterozygous Pfn1 KO at the embryonic level develop normally and do not exhibit any overt phenotype. I would recommend the authors to at least address this issue in the discussion.
- 3) Given that Pfn1 also interacts with microtubules (a critical player for mitosis) utilizing residues that are distinct from those required for actin and poly-proline-binding, it would be important to perform rescue experiments in KO cells with either wild-type or mutants of Pfn1 that are impaired in binding to various ligands and assess some of the mitosis and chromosomal phenotypes.

Reviewer #2 (Remarks to the Author):

Scotto di Carlo et al. studied the role of PFN1, a small actin binding protein previously implicated in a specific type of osteosarcoma (secondary to Paget's disease – OS/PDB), in maintaining accurate cell division. In this work, the authors provide evidence that PFN1 localizes to the midzone during anaphase, and that lack, or reduced levels, of PFN1 result in diminished actin polymerization at the cleavage furrow. This, in turn, leads to prolonged mitosis and higher frequency of chromosome missegregation events that result in micronuclei formation. Defective mitoses were also observed in a newly established Pfn1 knock-in mouse model. Finally, the authors demonstrate that Pfn1^{+/-} mouse fibroblast clones exhibit chromosome copy number alterations, and that OS/PDB human patients exhibit complex genome rearrangements, including chromothripsis, and frequent LOH of PFN1. The manuscript is well-written and easy to follow, and the points made are clear. The experiments were elegantly executed and the results support the main claims of the authors, i.e., that PFN1 has an important role in maintaining chromosome stability. The results would be of high interest to the field of cancer biology in general, and the chromosome instability field in particular. It is also interesting to see how an actin binding protein participates in proper cell division. Overall, the work is of high quality, but it would be important that the authors address the following comments before publication:

1. It is clear that reduced PFN1 results in lowered actin in the cleavage furrow, and increased chromosome instability. What is the effect of PFN1 overexpression in a wild-type background, and could PFN1 deficiency be corrected by re-expressing PFN1 (for example by transfection)?
2. The localization of PFN1 during mitosis was shown only in RPE1 cells. It would be interesting to see the location of PFN1 in additional human cells, as well as the location of Pfn1 in the mouse cells used in the study.
3. Figure 5i,j shows increased p53 in Pfn1^{+/-} mouse cells. This is under chronic conditions where Pfn1 levels are constitutively reduced. What would be the effect on p53 immediately after reduction of Pfn1 (for example in the shRNA RPE-1 cells).
4. If Pfn1 reduction results in aberrant rounding of mitotic cells, perhaps the authors could examine the morphology of mitotic cells when grown on different substrates (matrigel, or in low adherence

plates, for example). Is the effect growth substrate dependent?

5. If Pfn1 levels affect genome doubling, could the authors measure the levels of tetraploidy in the different culture systems (using a simple PI staining in FACS, compared to a diploid control)?

Minor comments:

1. Perhaps I missed it, but the frequency of micronuclei formation in +/+, +/-, -/- PFN1 RPE-1 cells is not shown.

2. Line 174: "suggesting that a p53-deficient background would enhance mitotic dysfunction and tumorigenicity". "would" should be replaced with "could".

3. In general, why not examine the role of Pfn1 reduction on human fibroblasts – a more relevant model to OS than RPE-1?

4. Line 218: instead of "replication" should be "mitotic/division".

5. The level of chromosome instability in mouse tissues from Pfn1+/- would be more relevant than the measurements (low pass sequencing) performed on cultured mouse clones, as they would better reflect the effect of reduced Pfn1 on mitotic fidelity in vivo. One potential approach could be isolation of single cells from Pfn1+/- and Pfn1 +/+ mouse tissues, performing single cell DNA sequencing to score for chromosome copy number changes, and comparing the wild-type with the heterozygote. This would, obviously, require more efforts, and so if not included in the current work, the authors should consider adding such an experiment to future studies, to substantiate the Pfn1+/- mice as a CIN model. Nevertheless, the authors should include WT controls to their analysis in supplementary figure 9, and indicate the level of gain/loss of each chromosome.

6. Line 248: the authors refer to high level copy number amplifications but do not provide further details explaining what genes were amplified, and the level of amplification.

7. In the discussion, line 271: "We used both CRISPR/Cas9 and shRNA experiments to inactivate PFN1 expression in RPE1 cells, which are widely used to study mitotic defects as well as chromosomal rearrangements in a p53-deficient background". This could confuse the reader into thinking that p53-/- cells were used in this study as well (which were not).

8. In Figure 5 KO should be HET in the different panels as these are Pfn1+/- derived.

9. In supplementary Figure 5 there is no indication for % of micronuclei, although the text suggests there is.

10. The relevance of the OD/PDB patient sequencing data is not immediately clear. There is no clear comparison between samples that have PFN1 LOH and ones that do not. Could this analysis be extended using already published sequences from human cancers? It would be helpful if the authors could provide more context and explain what can be learned from this sequencing efforts with regard to PFN1 LOH. Also, the ability to determine WGD relative to PFN1 LOH is interesting, and I would encourage making this analysis more accessible to the non-computational reader (for example, "WGD relative timing" is not a clear parameter).

11. The use of the phrase "nuclear shape" can mislead to think there is something wrong with the nuclear lamina, or membrane, when in fact the authors refer to micronuclei which are separate from primary nuclei. I recommend not using this terminology.

12. It would be interesting to look at the actin filaments in mitotic PFN1+/- (compared to WT) cells under an electron microscope.

Response to Reviewers

Reviewers' comments:

Reviewer #1 (Remarks to the Author):

This is an elegant study which performs a detailed mitotic characterization of in vitro and in vivo phenotypes of Pfn1 heterozygous and homozygous knockout cells demonstrating that loss of Pfn1 induces anaphase bridges, multipolar spindles, defects in cytokinesis and genome arrangements. While Pfn1's requirement for cleavage furrow formation and cytokinesis has been previously shown by another group (Bottcher et al. EMBO 2009), chromosomal changes upon loss of Pfn1 as demonstrated herein is novel. The paper is well-written and the conclusions are well-supported by experimental data. I have a few suggestions that would further improve the quality of the paper:

Thank you for your comment and the valuable feedbacks provided. We welcomed your suggestions. Please find below our detailed responses:

- 1. I would generally recommend WT-Pfn1-associated data next to KO phenotype data for comparison in all figures (the authors have done so in some but not for all the figures).**

*Thank you for pointing this out. Control images were missing in Figures 4, 5, and 6. We added WT-associated data in **Figure 6**, while for the other figures we added in the legend that control references could be found within other figures. For example, we added in the legend to Figure 4 the following sentence: “**Control metaphase, anaphase, and telophase images of dividing RPE1 cells are illustrated in Figure 1.**”*

- 2. Interestingly, some of the metrics (including CIN) seem to be comparable between Pfn1 heterozygous and homozygous knockout cells – yet there is a wide range of Pfn1 expression between cell types and importantly mice with heterozygous Pfn1 KO at the embryonic level develop normally and do not exhibit any overt phenotype. I would recommend the authors to at least address this issue in the discussion.**

We very much appreciate the Reviewer's observation and we have addressed this consideration in the Discussion. We have mentioned that homozygous knock-out of PFN1 can be achieved only in an immortalised background (RPE1 cells), while it is lethal in primary cells and in mice. However, we have stressed that the frequency of mitotic defects is comparable between heterozygous and homozygous KO cells, and this underlines that the loss of a single PFN1 copy is sufficient to drive errors. We have also added that, although adult heterozygous mice do not develop any overt phenotype, we noted a lower ratio of viable heterozygous knock-in to wild type mice, which suggests that some aberrant mutant embryos could be selected during development. Anyway, this is a different project that we aim at investigating in the next future, as also suggested by the Reviewer 2.

3. **Given that Pfn1 also interacts with microtubules (a critical player for mitosis) utilizing residues that are distinct from those required for actin and poly-proline-binding, it would be important to perform rescue experiments in KO cells with either wild-type or mutants of Pfn1 that are impaired in binding to various ligands and assess some of the mitosis and chromosomal phenotypes.**

We think this is an excellent suggestion and, in the attempt to address the Reviewer's concern, we have uncovered unexpected results about Profilin 1, which we summarise here. We have generated stable cell clones of WT, PFN1^{+/-}, and PFN1^{-/-} RPE1, each overexpressing either PFN1^{WT}, or the actin binding mutant PFN1^{R89E}, or the microtubule binding mutant PFN1^{G118V}. Mutations have been inserted through site-directed mutagenesis and then confirmed by Sanger sequencing. In order to verify whether and which of Profilin 1 mutants was able to rescue the defective mitoses, we performed time lapse experiments, and assessed frequency and duration of successful mitoses. We aimed at comparing the results of overexpression of mutants with those deriving from overexpression of wild type.

However, we surprisingly observed that overexpression of wild type Profilin 1 in KO cells failed to rescue the phenotype. Similarly, overexpression of neither PFN1^{R89E} or PFN1^{G118V} was able to reduce mitotic duration in KO cells. Moreover, we even noted that Profilin 1 overexpression in WT cells resulted in significantly prolonged mitosis, suggesting that increased actin polymerisation has the same effect than reduced actin polymerisation on mitotic progression. Although novel with respect of Profilin 1, this result is in agreement with activating mutations in other actin-promoting proteins (e.g., DIAPH1, WASp), which result in delayed mitoses. Therefore, a cell model of Profilin 1 deficiency (and hence, too little actin) and a cell model of Profilin 1 overexpression (and hence, too much actin) do lead eventually to the same phenotype of prolonged mitosis. In fact, overexpression of the actin-binding mutant (PFN1^{R89E}) did not delay cell division in WT cells.

*Concerning the microtubule binding mutant, we observed that, unlike PFN1^{R89E}, the PFN1^{G118V} protein was less stable and underwent degradation (in agreement with what we previously showed in Scotto di Carlo et al., JBMR 2020) (see **Supplementary Figure 12**). Overexpression of PFN1^{G118V} did not affect the duration of mitoses in any cell context. However, we cannot exclude that the lack of any effect could be due to protein degradation.*

Either way, we truly thank the Reviewer for this suggestion because it allowed us to uncover for the first time that both Profilin 1 overexpression and deficiency affect mitotic progression. We believe that this is a valuable improvement to our paper. Below, we report a bar graph showing our results, which were summarised in the manuscript as follows:

“Surprisingly, stable overexpression of Profilin 1 in PFN1^{+/-} and PFN1^{-/-} RPE1 cells failed to rescue a normal mitotic duration. On the contrary, its overexpression delayed mitotic progression in WT cells, which spent an average of 27.1 minutes (95% CI: 25.9 – 28.3) in mitosis versus 18.5 minutes of parental WT cells, suggesting that excess in actin nucleation likewise results in prolonged mitosis. This result is in agreement with activating mutations in other actin-promoting proteins⁵²⁻⁵⁴, which lead to delayed mitoses. Accordingly, stable overexpression of the

actin monomer-binding Profilin 1 mutant (PFN1^{R89E}; Ref⁵⁵) was not associated with mitotic delay in WT cells, and did not worsen the delayed phenotype in KO cells. Thus, balanced levels of Profilin 1, and therefore of actin, are necessary to modulate the kinetics of mitosis and to ensure a timely mitotic execution.”

Time taken from start (cell rounding) to end (cytokinesis) of mitosis; data are shown as mean \pm s.e.m.; (n mitoses = 77 WT, 42 WT+PFN1^{WT}, 19 PFN1^{-/-}, 64 PFN1^{-/-}+PFN1^{WT}, 25 PFN1^{-/-}, and 45 PFN1^{-/-}+PFN1^{WT} RPE1 cells). Data were analysed by Two-way ANOVA.

Reviewer #2 (Remarks to the Author):

Scotto di Carlo et al. studied the role of PFN1, a small actin binding protein previously implicated in a specific type of osteosarcoma (secondary to Paget’s disease – OS/PDB), in maintaining accurate cell division. In this work, the authors provide evidence that PFN1 localizes to the midzone during anaphase, and that lack, or reduced levels, of PFN1 result in diminished actin polymerization at the cleavage furrow. This, in turn, leads to prolonged mitosis and higher frequency of chromosome missegregation events that result in micronuclei formation. Defective mitoses were also observed in a newly established Pfn1 knock-in mouse model. Finally, the authors demonstrate that Pfn1^{+/-} mouse fibroblast clones exhibit chromosome copy number alterations, and that OS/PDB human patients exhibit complex genome rearrangements, including chromothripsis, and frequent LOH of PFN1.

The manuscript is well-written and easy to follow, and the points made are clear. The experiments were elegantly executed and the results support the main claims of the authors, i.e., that PFN1 has an important role in maintaining chromosome stability. The results would be of high interest to the field of cancer biology in general, and the chromosome instability field in particular. It is also interesting to see how an actin binding protein participates in proper cell division. Overall,

the work is of high quality, but it would be important that the authors address the following comments before publication:

We sincerely thank the Reviewer, we found all comments extremely helpful and have revised the manuscript accordingly. Please find below our responses:

- 1. It is clear that reduced PFN1 results in lowered actin in the cleavage furrow, and increased chromosome instability. What is the effect of PFN1 overexpression in a wild-type background, and could PFN1 deficiency be corrected by re-expressing PFN1 (for example by transfection)?**

We thank the Reviewer for this suggestion. Both Reviewers asked for rescue experiments with Profilin 1 overexpression, therefore we merged our response. We have generated stable cell clones of WT, PFN1^{+/-}, and PFN1^{-/-} RPE1, each overexpressing either PFN1^{WT}, or the actin binding mutant PFN1^{R89E}, or the microtubule binding mutant PFN1^{G118V}, these latter as required by the Reviewer 1. Mutations have been inserted through site-directed mutagenesis and then confirmed by Sanger sequencing.

To address the Reviewer's request, we first overexpressed Profilin 1 in WT cells, and interestingly noted that its overexpression resulted in significantly prolonged mitosis, suggesting that increased actin polymerisation has the same effect than reduced actin polymerisation on mitotic progression. Although novel with respect of Profilin 1, this result is in agreement with activating mutations in other actin-promoting proteins (e.g., DIAPH1, WASp), which result in delayed mitoses. Therefore, a cell model of Profilin 1 deficiency (and hence, too little actin) and a cell model of Profilin 1 overexpression (and hence, too much actin) do lead eventually to the same phenotype of prolonged mitosis. In fact, overexpression of the actin-binding mutant (PFN1^{R89E}) did not delay cell division in WT cells. With this novel notion in mind, we overexpressed wild type Profilin 1 in KO cells, thus switching from a model of actin deficiency to one of actin excess. And in fact, we found that re-expressing PFN1 in KO cells did not correct the phenotype.

We truly thank both Reviewers for this suggestion because it allowed us to uncover for the first time that both Profilin 1 overexpression and deficiency affect mitotic progression: too much or too little actin delay cell division. We believe that this is a valuable improvement to our paper. Below, we report a bar graph showing our results, which were summarised in the manuscript as follows:

Time taken from start (cell rounding) to end (cytokinesis) of mitosis; data are shown as mean \pm s.e.m.; (n mitoses = 77 WT, 42 WT+PFN1^{WT}, 19 PFN1^{+/-}, 64 PFN1^{+/-}+PFN1^{WT}, 25 PFN1^{-/-}, and 45 PFN1^{-/-}+PFN1^{WT} RPE1 cells). Data were analysed by Two-way ANOVA.

“Surprisingly, stable overexpression of Profilin 1 in PFN1^{+/-} and PFN1^{-/-} RPE1 cells failed to rescue a normal mitotic duration. On the contrary, its overexpression delayed mitotic progression in WT cells, which spent an average of 27.1 minutes (95% CI: 25.9 – 28.3) in mitosis versus 18.5 minutes of parental WT cells, suggesting that excess in actin nucleation likewise results in prolonged mitosis. This result is in agreement with activating mutations in other actin-promoting proteins^{52–54}, which lead to delayed mitoses. Accordingly, stable overexpression of the actin monomer-binding Profilin 1 mutant (PFN1^{R89E}; Ref⁵⁵) was not associated with mitotic delay in WT cells, and did not worsen the delayed phenotype in KO cells. Thus, balanced levels of Profilin 1, and therefore of actin, are necessary to modulate the kinetics of mitosis and to ensure a timely mitotic execution.”

2. **The localization of PFN1 during mitosis was shown only in RPE1 cells. It would be interesting to see the location of PFN1 in additional human cells, as well as the location of Pfn1 in the mouse cells used in the study.**

*We gladly accepted the Reviewer’s comment and performed Profilin 1 immunofluorescence in additional non-transformed cell lines, including the MC3T3 cells (used in this study), human dermal fibroblast derived from a control individual, and HK-2 cells (human kidney cells). We interestingly confirmed the enrichment of Profilin 1 in the spindle midzone of these cell lines, using two different antibodies, and hence modified the Results and the Methods sections accordingly. These images have been included as new **Supplementary Figure 2**.*

3. **Figure 5i,j shows increased p53 in Pfn1^{+/-} mouse cells. This is under chronic conditions where Pfn1 levels are constitutively reduced. What would be the effect on p53 immediately after reduction of Pfn1 (for example in the shRNA RPE-1 cells).**

*This is an interesting point raised by the Reviewer. To address it, we generated RPE1 cells stably expressing a PFN1-specific shRNA under a doxycycline inducible promoter. This new cell model, unlike the CRISPR and shRNA clones already described in the manuscript, allowed us to study the immediate effects of Profilin 1 reduction. Because Profilin 1 is a highly abundant protein, almost complete silencing of Profilin 1 (by 90%) is achieved after 7 days of doxycycline treatment. We show that, in this condition, p53 is not detected, indicating that p53 activation is not an immediate consequence of Profilin 1 reduction. However, a longer treatment (for 21 days), mimicking the chronic PFN1 inactivation, resulted in moderate p53 accumulation, suggesting that p53 is activated when Profilin 1 is constitutively reduced. We found this comment extremely constructive and very gladly included the results as new **Supplementary Figure 8**. We have therefore modified Results, Methods, and Discussion accordingly.*

- 4. If Pfn1 reduction results in aberrant rounding of mitotic cells, perhaps the authors could examine the morphology of mitotic cells when grown on different substrates (matrigel, or in low adherence plates, for example). Is the effect growth substrate dependent?**

To address this comment, we tried to culture RPE1 cells on low adherent plates as suggested by the Reviewer. However, this culturing method completely inhibited cell attachment, resulting in round-shaped suspension cells that could not be analysed. Because in vivo mitotic cells generate protrusive forces that deform the surrounding collagen fibres of the extracellular matrix to allow for mitotic rounding, we decide to verify the substrate-dependent growth on collagen-coated coverslips. However, we did not observe any changes in the frequency of aberrant mitotic morphology of PFN1-KO cells when grown on collagen. Below, we report the graph of the results obtained, which were instead summarised in the manuscript as follows:

“Because in vivo mitotic cells generate protrusive forces that physically deform the surrounding collagen fibres of the extracellular matrix to allow for mitotic rounding and elongation^{40,50,51}, we suspected that defective rounding might be rescued by growth on collagen plates. However, growth on collagen-coated coverslips failed to restore the spherical geometry of PFN1-KO mitotic cells (data not shown), indicating that the observed effect is not substrate-dependent.”

Quantification (%) of non-spherical metaphase cells in WT, *PFN1*^{+/-}, and *PFN1*^{-/-} RPE1 cells after thymidine-nocodazole block ($n = 206$ WT, 260 *PFN1*^{+/-}, and 139 *PFN1*^{-/-} RPE1 cells on uncoated coverslips, and 244 WT, 238 *PFN1*^{+/-}, and 177 *PFN1*^{-/-} RPE1 cells on collagen-coated coverslips). ns WT = 0,9810; ns *PFN1*^{+/-} = 0,9756; ns *PFN1*^{-/-} = 0,5129. Data were analysed by Two-way ANOVA.

We added the detailed methods of cell culture on collagen-coated coverslips and scoring of aberrant mitotic rounding in the Methods section.

5. If Pfn1 levels affect genome doubling, could the authors measure the levels of tetraploidy in the different culture systems (using a simple PI staining in FACS, compared to a diploid control)?

*To measure the levels of tetraploidy in Profilin 1-deficient cells, we analysed the ploidy content of *PFN1*^{-/-} RPE1 clones compared with WT cells, through FACS analysis of propidium iodide incorporation. To discriminate between tetraploid peaks and cells with duplicated DNA (G2/M), cells were serum-starved in G0/G1 phase of cell cycle. To accurately identify the G0/G1 diploid peak position, we used DNA from human peripheral blood mononuclear cells (PBMCs) as a diploid internal standard. We found that $4.5 \pm 0.3\%$ of *PFN1*-null cells were tetraploid, which is quite in agreement with the frequency of cytokinesis failures detected. We included this result as **Supplementary Figure 3b** (previously, Supplementary Figure 2). To further exclude that those could represent KO cells that escaped serum starvation, aliquots of the same cell populations were every time subjected to immunofluorescence detection of phospho-histone H3, a mitotic marker. We detected a definitely negligible amount of pH3-positive cells, which is comparable to what observed in wild type cells as 4n peak.*

Minor comments:

1. Perhaps I missed it, but the frequency of micronuclei formation in +/+, +/-, -/- PFN1 RPE-1 cells is not shown.

*Thank you for pointing this out. The reviewer is correct, and we have added the bar graph showing the % of micronuclei in *PFN1*^{+/+}, *PFN1*^{+/-}, and *PFN1*^{-/-} RPE1 cells in **Supplementary Figure 6** (previously, Sup. Fig. 5).*

2. **Line 174: “suggesting that a p53-deficient background would enhance mitotic dysfunction and tumorigenicity”. “would” should be replaced with “could”.**
We agree and we have revised it accordingly.
3. **In general, why not examine the role of Pfn1 reduction on human fibroblasts – a more relevant model to OS than RPE-1?**
We completely agree with the Reviewer. We also thought of fibroblast as cell model for our study in the first place, especially patient-derived fibroblasts harbouring the PFN1 mutation. However, we tried to reach out PFN1-mutated patients from the pedigree described in Scotto di Carlo et al., JBMR 2020, but none of them were available for a skin biopsy. We preferred not to use CRISPR/Cas9- or shRNA-mediated inactivation because it would have required too many passages for clone selection, which fibroblasts would not tolerate as primary cells. For all these reasons, we decided to use MC3T3 in this study, because, albeit murine, they are mesenchymal like osteosarcoma cells.
4. **Line 218: instead of “replication” should be “mitotic/division”.**
We agree and we have revised it accordingly.
5. **The level of chromosome instability in mouse tissues from Pfn1+/- would be more relevant than the measurements (low pass sequencing) performed on cultured mouse clones, as they would better reflect the effect of reduced Pfn1 on mitotic fidelity in vivo. One potential approach could be isolation of single cells from Pfn1+/- and Pfn1 +/+ mouse tissues, performing single cell DNA sequencing to score for chromosome copy number changes, and comparing the wild-type with the heterozygote. This would, obviously, require more efforts, and so if not included in the current work, the authors should consider adding such an experiment to future studies, to substantiate the Pfn1+/- mice as a CIN model.**
Thank you for this suggestion. We find it very helpful. However, as also supposed by the Reviewer him/herself, it was not feasible within the timeframe of the current revision. We are planning to include it as a different project in the next future, which will include the complete characterisation of the Pfn1 mouse model.
Nevertheless, the authors should include WT controls to their analysis in supplementary figure 9, and indicate the level of gain/loss of each chromosome.
To address the Reviewer’s concern, we have updated the Supplementary Figure 9 (now Supplementary Figure 11) to show the magnitude of copy number gains and losses of each chromosome relative to a wild type sample. However, a control sample cannot be added in this figure because the copy number plot shows normalised data of KO clones compared with WT control, which was therefore subtracted as baseline.
6. **Line 248: the authors refer to high level copy number amplifications but do not provide further details explaining what genes were amplified, and the level of amplification.**
We have not provided details about copy number amplifications in the OS/PDB samples because CNV frequently did not contain genes, but rather affected intergenic regions. In the case where genes were involved, we did not notice gene recurrency

among specimens, likely due to a low sample size. The levels of copy number amplifications are shown in Figure 7a for the WGS samples, while the exact level of amplification for PFN1 in all samples is shown in the Supplementary Table 2. The fraction of the genome affected by copy number events is shown in Figures 7b-c.

7. **In the discussion, line 271: “We used both CRISPR/Cas9 and shRNA experiments to inactivate PFN1 expression in RPE1 cells, which are widely used to study mitotic defects as well as chromosomal rearrangements in a p53-deficient background”. This could confuse the reader into thinking that p53-/- cells were used in this study as well (which were not).**

*Thank you for noticing and allowing us to clarify this point. We have revised the sentence, which now reads as follows: “We used both CRISPR/Cas9 and shRNA experiments to inactivate PFN1 expression in RPE1 cells, which are widely used to study mitotic defects and chromosomal rearrangements, **although in a p53-deficient background**^{12,69-71}, which was not needed in our study.”*

8. **In Figure 5 KO should be HET in the different panels as these are Pfn1+/- derived.**

We agree with the Reviewer and replaced “KO” with “Pfn1^{+/-}” in panels i, j, and k of Figure 5.

9. **In supplementary Figure 5 there is no indication for % of micronuclei, although the text suggests there is.**

*As for the minor comment 1, we have added the bar graph showing the % of micronuclei in PFN1^{+/+}, PFN1^{+/-}, and PFN1^{-/-} RPE1 cells in **Supplementary Figure 6** (previously, Sup. Fig. 5).*

10. **The relevance of the OD/PDB patient sequencing data is not immediately clear. There is no clear comparison between samples that have PFN1 LOH and ones that do not. Could this analysis be extended using already published sequences from human cancers? It would be helpful if the authors could provide more context and explain what can be learned from this sequencing efforts with regard to PFN1 LOH. Also, the ability to determine WGD relative to PFN1 LOH is interesting, and I would encourage making this analysis more accessible to the non-computational reader (for example, “WGD relative timing” is not a clear parameter).**

We thank the Reviewer for pointing out this issue. We have now specified which samples have loss of heterozygosity of PFN1 in Supplementary Table 2 (please, see column H “PFN1 LOH”). We have also modified the main text to explain the intuition underpinning the timing analysis of WGD and make that analysis more accessible to non-expert readers. We decided to focus on osteosarcoma genomes to assess the relevance of PFN1 inactivation in human cancers due to our previous work reporting loss-of-function mutations in early onset Paget's disease of bone. In these tumours, we observe loss of heterozygosity at the PFN1 locus and multiple copies of PFN1. We believe that such configuration is more likely to occur through loss of one copy of PFN1, then followed by WGD (as we observe in vitro) and therefore duplication of the

same parental allele, rather than through WGD followed by loss of multiple PFN1 copies originating from the same paternal allele. We have now extended the presentation of these results in the main text. The current analysis cannot be extended using already published sequences from human cancers because the genomic involvement of PFN1 in other cancer types is too low to be analysed in comparative studies. Please, see below the PFN1 pancancer plot, showing little genetic affection of the gene in multiple cancers.

11. The use of the phrase “nuclear shape” can mislead to think there is something wrong with the nuclear lamina, or membrane, when in fact the authors refer to micronuclei which are separate from primary nuclei. I recommend not using this terminology.

We agree and we have revised it accordingly.

12. It would be interesting to look at the actin filaments in mitotic PFN1+/- (compared to WT) cells under an electron microscope.

Thank you for this suggestion. It would have been interesting to explore this aspect. However, this was not possible in the current study: to examine actin organization with single filament resolution, we should use platinum replica electron microscopy, which is an expertise that we do not have. Unfortunately, setting up the protocol would have required more than the 6 months provided for the revision. We hope that the Reviewer understands our difficulty in addressing this comment.

REVIEWERS' COMMENTS:

Reviewer #1 (Remarks to the Author):

The authors have satisfactorily addressed my previous comments

Reviewer #2 (Remarks to the Author):

Dear authors and editors,

I found the revision fully satisfactory. I was happy to see that the rescue experiment performed contributed to our understanding of Pfn1 biology. Perhaps I missed it, but did the authors include the figure presented in the letter to the manuscript itself? I would recommend doing so.

I have no other comments and recommend accepting this manuscript for publication.